



# Use of filter radiometer measurements to derive local photolysis rates and for future monitoring network application

Hannah L. Walker[1,2], Mathew R. Heal[1], Christine F. Braban[2], Mhairi Coyle[2,3], Sarah R. Leeson[2], Ivan Simmons[2], Matthew R. Jones[2], Richard Kift[4], and Marsailidh M. Twigg[2]

[1]School of Chemistry, University of Edinburgh, David Brewster Road, Edinburgh, EH9 3FJ, UK
[2]UK Centre for Ecology & Hydrology, Bush Estate, Penicuik, Edinburgh, EH26 0QB, UK
[3]The James Hutton Institute, Craigiebuckler, Aberdeen, AB15 8QH, UK
[4]Department of Earth and Environmental Sciences, University of Manchester, Oxford Road, Manchester, M13 9PL, UK

**Correspondence:** H. L. Walker (hannah.walker@ed.ac.uk) and M. R. Heal (m.heal@ed.ac.uk)

**Abstract.** Production of hydroxyl (OH) radicals is frequently dominated by the photolysis of tropospheric ozone ($O_3$). How-ever, photolysis of nocturnal radical reservoirs, such as nitrous acid (HONO) and nitryl chloride ($ClNO_2$), also produces radicals (OH and Cl atoms) that contribute to the oxidising capacity of the local atmosphere, and initiate many radical-chain reactions that lead to the formation of harmful secondary pollutants. Photolysis of nitric acid ($HNO_3$) is also a minor radical

production mechanism. In this paper, locally representative photolysis rate constants (*j*-values) for these molecules are shown to be critical for quantifying and understanding the rate of radical production in a local atmosphere.

The first long-term 4-$\pi$ filter radiometer dataset in the UK (21 November 2018-20 November 2019) available for direct atmospheric model validation is reported. Measurements were made at Auchencorth Moss, a Scottish rural background site, and $j(NO_2)$ is used to generate a measurement-driven adjustment factor (MDAF) for calculated *j*-values that accounts for local

changes in meteorological variables without significantly increasing computational cost.

Modelled clear-sky *j*-values and actinic flux for Auchencorth Moss were generated using the Tropospheric Ultraviolet and Visible radiation model (TUV; v.5.3.1). Applying the MDAF metric resulted in the calculated photolytic production rate of OH radicals, from all sources considered, being $\sim 40\%$ lower over the year. Photolysis of HONO resulted in an increased rate of OH production compared to that from $O_3$ in low-light conditions, such as sunrise and sunset (Solar Zenith Angle

>80°). Hydroxyl radical production from HONO photolysis exceeded that from $O_3$ consistently throughout the day during the winter and autumn (by a factor of 5 and 2.1, respectively). Radical production rates from HONO and $ClNO_2$ reached maximum values during the early morning hours of summer (06:00-09:00 UTC), with OH produced at a rate of $1.03 \times 10^6$ $OH\,radicals\,cm^{-3}\,s^{-1}$, and Cl radicals at $3.20 \times 10^4\,Cl\,radicals\,cm^{-3}\,s^{-1}$, with the MDAF metric applied.

This first application of the MDAF *j*-values demonstrates an efficient measurement and computational approach to im-

prove modelling of the local atmospheric photochemistry that drives $NO_2$, $O_3$ and PM pollution levels. The incorporation of local radiation measurements in measurement networks, and the consequent greater spatial resolution of locally-relevant photolysis coefficients in model photolysis parameterisations, will improve the accuracy of assessment of air pollution and policy-intervention impacts.





## 1 Introduction

Atmospheric chemistry is largely driven by solar radiation. The photolysis of nitrogen dioxide ($NO_2$) has significant impact on air quality, contributing to the formation of harmful pollutants like tropospheric ozone ($O_3$; Reactions R1 and R2).

$$NO_2 + h\nu\,(\lambda \leq 420\,nm) \quad \rightarrow \quad NO + O(^3P) \tag{R1}$$
$$O(^3P) + O_2 + M \quad \rightarrow \quad O_3 + M \tag{R2}$$

Photodissociation of many trace gases produce highly reactive radical species such as hydroxyl (OH) and chlorine atoms (Cl), which dominate the oxidising capacity of the atmosphere. Since reactions with these radicals are often the rate-determining step in the chemical loss of many species emitted into the atmosphere, such as volatile organic compounds (VOCs), these radicals dictate the atmospheric lifetimes and concentrations of these other species (Atkinson and Arey, 2003). Products of these reactions include a variety of harmful secondary pollutants like tropospheric ozone ($O_3$), and atmospheric reservoirs of $NO_2$ such as peroxyacetyl nitrate (PAN), that contribute to long-range and regional pollution (Spataro and Ianniello, 2014). Conversely, the high reactivity of these radicals means they have very short atmospheric lifetimes (e.g. <1 s for OH; Monks, 2005), so their concentrations and potential oxidation impacts are only relevant on a local scale.

The OH radical is the most dominant radical diurnally, known to be produced through the photolysis of ozone and subsequent reaction with $H_2O$ (Reactions R3 and R4).

$$O_3 + h\nu\,(\lambda \leq 340\,nm) \quad \rightarrow \quad O(^1D) + O_2 \tag{R3}$$
$$O(^1D) + H_2O \quad \rightarrow \quad 2OH \tag{R4}$$

Photolysis of oxidised reactive nitrogen species ($NO_y$ = sum of $NO_x$, HONO, $HNO_3$, $2N_2O_5$, $ClNO_2$ and organic gases with $NO_x$ groups) also play an important role in radical formation. Nitrous acid (HONO) has been attributed as a large source of OH radicals in multiple locations across the globe (Reaction R5; Alicke et al., 2003; Elshorbany et al., 2009; Villena et al., 2011; Ryan et al., 2018), while nitryl chloride ($ClNO_2$) is the dominant early morning source of chlorine atoms (Reaction R6; Young et al., 2012, 2014; Riedel et al., 2014).

$$HONO + h\nu\,(305\,nm < \lambda < 420\,nm) \quad \rightarrow \quad OH + NO \tag{R5}$$
$$ClNO_2 + h\nu\,(\lambda \leq 475\,nm) \quad \rightarrow \quad Cl + NO_2 \tag{R6}$$

Production of Cl radicals from measured levels of $ClNO_2$ were first estimated in coastal locations near Houston, Texas (Osthoff et al., 2008), where an abundance of aerosol chloride in the form of sea salt coexists with nitrogen oxides ($NO_x$). However, formation of $ClNO_2$ is not limited to chloride in sea spray. Its presence has been repeatedly measured at sites up to 1400 km from the nearest coastline (Thornton et al., 2010; Bannan et al., 2015; Osthoff et al., 2018; Sommariva et al., 2018), with the particulate chloride precursor generated from anthropogenic sources such as fossil fuel combustion and biomass burning (Ahern et al., 2018).




Cl radicals are extremely reactive with VOCs, with rate constants often an order of magnitude greater than equivalent reactions with OH (Monks, 2005; Young et al., 2012). Therefore, even at low concentrations, Cl has a significant effect on local tropospheric oxidation (Riedel et al., 2014; Bannan et al., 2015). As a result, quantifying local photolysis rates is as critical to understanding the local air quality as measuring trace gas concentrations.

Photolysis is a unimolecular reaction characterised by a simple rate equation, demonstrated by Eq. (1) for the photolysis of
$NO_2$ in Reaction (R1).

$$-\frac{d[NO_2]}{dt} = j(NO_2) \times [NO_2] \tag{1}$$

The rate constant for this photolysis reaction, $j(NO_2)$, can be measured in absolute terms using a chemical actinometer. This estimates $j(NO_2)$ by measuring the concentration of NO following photodissociation of a known concentration of $NO_2$ inside a reaction chamber exposed to ambient light (Bahe et al., 1980; Shetter et al., 1992; Lantz et al., 1996; Bohn et al.,
2005). Spectroradiometers measure actinic flux independent of the angle of incidence in a 2-$\pi$ sr field of view, as a function of wavelength (Kraus and Hofzumahaus, 1998). Photolysis frequencies are determined from Eq. (2) for any molecule whose absorption cross-section ($\sigma$) falls within the measured wavelength ($\lambda$) range (Shetter and Müller, 1999). For instruments at ground level, the lower-bound wavelength limit ($\lambda_1$) is typically set to 290 nm, while the upper-bound is dependent on the molecule (e.g. $\lambda_2$ = 420 nm for $j(NO_2)$, and $\lambda_2$ = 340 nm for $j(O^1D)$; Edwards and Monks, 2003).

$$j(NO_2) = \int_{\lambda_1}^{\lambda_2} \sigma(\lambda, T)\, \phi(\lambda, T) F(\lambda)\, d\lambda \tag{2}$$

The actinic flux ($F$) is the quantity of light available to molecules at a given point in the atmosphere, that upon absorption results in photodissociation (Kylling et al., 2003). It is a spherical integration of solar radiant energy over a surface plane, differing by a cosine of the angle of incidence (Madronich, 1987). The absorption cross-section ($\sigma$) quantifies the ability of the molecule to absorb radiation, and the quantum yield ($\phi$) is the probability that the photodissociation leads to the product channel
under consideration, in this case $O(^3P)$ and NO. These are molecule-specific parameters that vary with respect to wavelength and temperature, the latter of which is not often assessed for molecules. Hence, uncertainty in these values propagates through to uncertainty in calculated photolysis frequencies (Shetter et al., 1996; Kraus and Hofzumahaus, 1998).

Filter radiometers use band-pass filters to measure broadband actinic flux, between $\lambda_2 - \lambda_1$ in Eq. (2), designed to simulate the $\sigma \times \phi$ product of the molecule of interest at its strongest absorbing region. A 4-$\pi$ sr filter radiometer provides a 360° field of
view of the surrounding environment by utilising two identical optical inlets, designed by Junkermann et al. (1989) and modified by Volz-Thomas et al. (1996). The two 2-$\pi$ sr domes face in opposite directions, separating the total measured actinic flux into down- and up-welling components ($F \downarrow$ and $F \uparrow$, respectively). These instruments exhibit good linear detector responses, and long-term stability in calibration factors (Bohn et al., 2008), and measurements can be recorded with high time resolution (1 s). These instruments are also easy to deploy and maintain, making them excellent candidates for routine measurements.
However they remain reliant on absolute calibrations to quantify $j(NO_2)$ from recorded voltages, and measurements are only applicable for specified reactions, with limited potential to estimate the photolysis frequencies of other atmospheric species.



Solar global irradiance ($G$) measurements are more widespread than $j(\mathrm{NO_2})$, as this is a common meteorological parameter included in multiple global monitoring networks, such as the World Meteorological Organisation (WMO) Global Atmospheric Watch (GAW) programme. Extensive efforts have been made to derive non-linear parameterisations that utilise existing measurements of $G$ to quantify downwelling $j(\mathrm{NO_2})$, thereby improving spatial coverage of $j$-value estimations (Bahe et al., 1980; Brauers and Hofzumahaus, 1992; Webb et al., 2002; Trebs et al., 2009). $G$ is typically measured using a pyranometer, which has a horizontal sensor measuring total downwelling radiation (direct + diffuse), weighted by a cosine response subject to the angle of incident light (Webb et al., 2002). Trebs et al. (2009) extensively discuss this relationship, and present good agreement between co-located $G$ and $j(\mathrm{NO_2})\downarrow$ measurements for 7 ground-based field sites up to 800 m above sea level despite measurements being from different years, seasons and continents. The authors acknowledge that this parameterisation is not a substitute for measurements of $j(\mathrm{NO_2})$, but a viable alternative to radiation transfer calculations where input parameters (like cloud cover) are inadequately known.

Most atmospheric and radiative transfer models use Eq. (2) to estimate $j$-values, but often oversimplify local meteorological conditions due to the computational cost of their inclusion, or the absence of necessary measured input parameters (Shetter et al., 2003). A myriad of variables impact the photolysis frequency, such as cloud cover, aerosol optical depth (AOD; Wild et al., 2000), surface albedo and ozone column. All of these variables can redirect incident radiation by absorption and/or reflection. Palancar et al. (2013) demonstrate that when measured AOD and $\mathrm{NO_2}$ concentrations are included in model inputs, actinic flux is reasonably well predicted and the main source of uncertainty is then attributable to $\sigma$ and $\phi$.

This study presents the first long-term filter radiometer dataset in the UK (Auchencorth Moss, SE Scotland) co-located with relevant ancillary measurements to quantify radical production rates from photolysis pathways. It aims to assess the temporal distribution of local OH and Cl radical formation at a single site, using the filter radiometer measurements applied to clear-sky model outputs to account for changes in local meteorological variables. The inclusion of local values of such variables would otherwise considerably increase model computational cost. The dataset provides the first accurate long-term UK point of comparison for validation of model photolysis coefficients. The scientific, and ultimately policy, value of incorporating filter radiometer measurements alongside existing long-term and spatially-resolved air quality monitoring networks is emphasised.

## 2 Method

### 2.1 Site description

Auchencorth Moss is a rural air quality monitoring station located in south-east Scotland, approximately 18 km SSW of Edinburgh (lat: $55°47'36''$ N, lon: $3°14'41''$ W, altitude: $260$ m). The site operates as a Level II/III European Monitoring and Evaluation Programme (EMEP) "supersite"), representative of the northern UK background atmosphere, at which atmospheric components are measured using standard methodology (Tørseth et al., 2012). Collectively, EMEP monitoring sites are used to understand long-range transport of air pollution around Europe, verify regional modelling approaches and advise governmental bodies on concentration and deposition of pollutants (UNECE, 2004; Aas et al., 2012).



Auchencorth Moss is also a site for a number of national and local monitoring networks (see Twigg et al., 2015), hosting an

extensive array of instrumentation to measure trace species concentrations and meteorological parameters (Coyle et al., 2019). Long-term instrumentation deployed at Auchencorth Moss utilised in this study is detailed in Table 1. It is acknowledged that both the HONO and $HNO_3$ concentrations reported by the Monitor for AeRosols and Gases in Ambient Air (MARGA; Metrohm Applikon, NL) have potential interferences from other $NO_y$ species. These interferences have been shown to result in both over- (Phillips et al., 2013) and underestimates (Rumsey et al., 2014) in reported concentration. Since these interferences

have not yet been formally quantified, this study uses the reported values as they are with caution.

## 2.2  Filter radiometer

The 4-$\pi$ filter radiometer (Metcon, Meteorologie Consult GmbH, DE) was mounted $\sim 3$ m above the ground, recording measurements at 1 s time resolution for a full year (21 November 2018–20 November 2019). The inlet optic of each dome is designed to have a near-uniform angular response through use of a quartz diffusor (Bohn et al., 2004). Each optical inlet is

surrounded by a light shield to provide an "artificial horizon", restricting the field-of-view for each dome to one hemisphere (Volz-Thomas et al., 1996) and preventing overlap. Transmitted light is guided through a set of optical filters (2-mm UG3, 1-mm UG5, Schott GmbH) that restrict transmitted photons to wavelengths of interest prior to their detection by a Hamamatsu photodiode, which proportionally converts incident radiation into an output voltage.

Output signals from the filter radiometer were calibrated against a Bentham DTM300 scanning spectroradiometer between

13 and 25 June 2019. This mid-summer period was selected to provide calibration over the maximum range of ambient incident radiation. The direction of the filter radiometer was turned $180°$ mid-way through this period, in order to calibrate each dome separately. $j(NO_2)$ was calculated from the actinic flux spectra measured by the spectroradiometer using $\sigma(NO_2)$ from Mérienne et al. (1995) and $\phi(O^3P)$ from Troe (2000). Calibration factors for each dome were quantified using data presented in Fig. 1.

The limits of detection (LOD; $3\sigma$ of background signal) were $9.40 \times 10^{-6}$ $s^{-1}$ and $1.15 \times 10^{-5}$ $s^{-1}$ for the down- and upwelling domes, respectively. Background signals were determined from averaged measurements made after sunset and before sunrise (solar zenith angle (SZA) $\geq 96°$) and removed prior to data analysis. On a few occasions of peak solar irradiance (noon in summer), recorded voltage of the downwelling dome exceeded the range of the detector and was reported as $\sim 10$ V, observed during the calibration in Fig. 1. These incidences of high $j(NO_2)\downarrow$ ($< 9$ V) comprised $1.4\%$ of all data collected,

and were removed from the dataset before further analysis. As a consequence, maximum $j(NO_2)\downarrow$ presented in this study is an underestimate ($7 - 22\%$ based on calibration). During the calibration, measurements made by the filter radiometer were averaged to match the duration of each spectroradiometer scan (3 mins). The standard deviation associated with each of the averaged filter radiometer measurements were used to remove calibration data points where actinic flux was highly variable, and not comparable between the instruments during the 3 min scan.

The uncertainty of the filter radiometer measurements were estimated as a combination of instrumental error (e.g. non-ideal inlet optic responses; Larason and Cromer, 2001), error from calibration, and errors due to external factors (e.g. temperature stability). For six of the same filter radiometers, Shetter et al. (2003) quoted overall error as $9.6 - 11\%$, the range dependent





on whether conditions were clear or cloudy. To provide a conservative estimate of error for both domes of the filter radiometer
used here, the upper bound (11%; cloudy conditions) is combined with the calibration error of each dome. Overall errors for
down- and upwelling domes were 13% and 12%, respectively.

## 2.3 Model approach

Calculations of photolysis frequencies for clear-sky conditions were performed using the Tropospheric Ultraviolet and Visible
radiation model (TUV v5.3; Madronich, 1987; NCAR, 2019), one of the most widely used radiation models (Lee-Taylor and
Madronich, 2002; Bais et al., 2003; Wilson, 2015; Ghosh and Sarkar, 2016; Wang et al., 2017; Xu et al., 2019). Photolysis
frequencies ($j$-values) were calculated for the photodissociation of $NO_2$, $O_3$, HONO, $HNO_3$ and $ClNO_2$. The model was set
up for the location and height of the filter radiometer at Auchencorth Moss. Variables were set to assume clear-sky conditions,
and those not influenced by cloud cover were kept constant for all calculations. Ozone column was estimated as the mean of
daily Dobson spectrophotometer measurements at Lerwick for the study duration (Defra, 2020b), while default TUV values
for surface albedo and AOD at $500\,\mathrm{nm}$ were used.

The model output included both actinic flux by wavelength ($290 - 420\,\mathrm{nm}$), and the $j$-values calculated for each hour of
the day. The latter were used to determine an hourly measurement-driven adjustment factor (MDAF), shown in Eq. (3), while
the former was used to determine $j(ClNO_2)$ using Eq. (2) and updated $\sigma(ClNO_2)$ measurements made by Ghosh et al. (2012),
now the preferred IUPAC values (as of June 2013; Atkinson et al., 2007). It should be noted that for $j(NO_2)$ calculations, the
molecule-specific parameters used by the TUV model are the same as those used in the calibration of the filter radiometer.

## 170 2.4 Theoretical calculations

### 2.4.1 Adjustment factor

Measurements of actinic flux made using radiometers intrinsically capture variations in solar flux caused by local meteorology
and other factors, such as the presence of aerosols. Since it is computationally intensive to replicate these variations in radiation,
which cannot be well represented if input parameters are poorly known, it has been demonstrated that radiometer measurements
could be used to adjust $j$-values in chemical transport models (Elshorbany et al., 2012; Sommariva et al., 2020). Equation (3)
illustrates how this can be done by defining a measurement-driven adjustment factor (MDAF), using $j(NO_2)$ as a reference.

$$MDAF = \frac{measured\ j(NO_2)}{modelled\ j(NO_2)} \tag{3}$$

Correction factors like the MDAF have only occasionally been applied both spatially and temporally (Stone et al., 2012;
Bannan et al., 2015), with measured and modelled environments often not co-located (e.g. Sommariva et al., 2020). The
MDAF metric, or equivalent, can only be used for model validation where radiometer measurements to complement the model
study can be obtained.





### 2.4.2 Production rate of OH radicals

The application of a locally-derived MDAF to model-calculated $j$-values is demonstrated here for calculations of the rate of production of OH radicals at Auchencorth Moss from $O_3$ and two $NO_y$ species, HONO and $HNO_3$. For both the latter, OH radicals are produced by a direct photodissociation as shown in Reaction (R5) for the case of HONO. The rates are calculated using Eq. (4), and an equivalent for the photolysis of $HNO_3$.

$$p(OH)_{HONO} = j(HONO) \times [HONO] \tag{4}$$

These are compared with the production rate from $O_3$ photolysis (Reactions R3 and R4) as calculated by Eq. (5) and (6).

$$p(OH)_{O_3} = 2 \times f \times j(O^1D) \times [O_3] \tag{5}$$

$$f = \frac{k_{H_2O}[H_2O]}{k_{N_2}[N_2] + k_{O_2}[O_2] + k_{H_2O}[H_2O]} \tag{6}$$

In these equations, $j(O^1D)$ is the photolysis rate constant for $O_3$, and $f$ is the fraction of $O(^1D)$ atoms that react with water vapour to form OH, as opposed to their quenched removal by $N_2$ and $O_2$ molecules. Local relative humidity and temperature measurements (Coyle et al., 2020) were used to derive absolute $H_2O$ concentrations, and hourly concentrations of $O_3$, HONO and $HNO_3$ measured at Auchencorth Moss were downloaded directly from UK-AIR (Defra, 2020a). For the study period at Auchencorth Moss, the fraction $f$ was $6.1 \pm 2\%$. It should be noted that while $O_3$ is reasonably well-mixed throughout the boundary layer, HONO has been reported to have a clear vertical profile where concentrations are strongly enhanced at ground level (Kleffmann et al., 2003; Young et al., 2012; Ryan et al., 2018). As HONO is measured at a height of 3.55 m at Auchencorth Moss (Twigg et al., 2015), $p(OH)_{HONO}$ presented here are applicable for ground level, and would be an overestimate for higher altitudes (>1 km).

### 2.4.3 Production rate of Cl atom radicals

The MDAF is also demonstrated for calculation of the production rate of Cl atoms from the photodissociation of $ClNO_2$, shown in Eq. (7), to the products in Reaction (R6).

$$p(Cl)_{ClNO_2} = j(ClNO_2) \times [ClNO_2] \tag{7}$$

Values of $\sigma(ClNO_2)$ used to determine $j(ClNO_2)$ were compared between those used in the TUV model (Sander et al., 2006) and the more recent IUPAC preferred values from Ghosh et al. (2012). Since the dataset presented by Ghosh et al. includes the temperature dependence of $\sigma(ClNO_2)$, values in this study were adjusted according to the daily mean temperature measured at Auchencorth Moss prior to further calculation. Hourly $j(ClNO_2)$ was then determined using Eq. (2) from the temperature adjusted $\sigma(ClNO_2)$ and hourly actinic flux from the TUV model. Henceforth, $j(ClNO_2)$ values calculated in this way are referred to as "updated $j(ClNO_2)$". The MDAF was applied to both the TUV and updated $j(ClNO_2)$ values, and the production rate of Cl radicals ($p(Cl)_{ClNO_2}$) using all four $j(ClNO_2)$ estimates are evaluated.

$ClNO_2$ is not routinely measured at Auchencorth Moss, so an estimate for its concentrations at this rural site was obtained from the literature. Sommariva et al. (2018) reported a median diurnal concentration of 21.5 ppt at Weybourne Atmospheric





Observatory, UK ($52°47'$ N, lon: $1°07'$ E) during the summer of 2015. This is a rural site on the coast of East England, whose concentrations of $ClNO_2$ are the best available measured proxy for Auchencorth Moss values. In the calculations of

$p(Cl)_{ClNO_2}$ presented here, 21.5 ppt ($0.074\,\mu g\,m^{-3}$, at Auchencorth Moss annual average $T$ and $p$) is used as both a constant concentration, and as the mean concentration of $ClNO_2$ for an approximate diurnal cycle that might be expected at Auchencorth Moss (Fig. 2). The shape of the diurnal profile follows that reported by Sommariva et al. (2018) at Weybourne.

## 3   Results and discussion

### 3.1   Concentrations of trace gases and meteorology

Figure 3 shows the time series of $O_3$, HONO and $HNO_3$ concentrations, total solar global irradiance ($G$), temperature, relative humidity, wind speed and total (downwelling + upwelling) $j(NO_2)$ for the year of measurements. Summary statistics of measured trace gas concentrations during each season are presented in Table 2, and diurnal cycles of $O_3$ and HONO for each season are shown in Fig. 4. UK seasons are defined by meteorological convention: winter (December to February), spring (March to May), summer (June to August) and autumn (September to November). All times are expressed in UTC.

Seasonal median $O_3$ concentrations ranged from $51.2\,\mu g\,m^{-3}$ in autumn, to $74.2\,\mu g\,m^{-3}$ in spring. The $O_3$ diurnal cycle is generally weak but most evident in spring and summer, where concentrations increase after sunrise and peak mid-afternoon at 15:00. Diurnal variation is also observed in both autumn and winter, but is far less pronounced. This is in part, due to high wind speeds at this rural site ($0.26 - 18.8\,m\,s^{-1}$, median of $5.0\,m\,s^{-1}$) which enhances vertical mixing. This is particularly evident at night where the boundary layer is generally shallower. The largest $O_3$ peaks are observed in April, May and August,

coincident with the spring-time annual maximum in $O_3$ in the UK and increases in ambient temperature.

Median HONO concentrations show far less diurnal and seasonal variation, but a greater interquartile range of concentrations for each hour in the summer. Larger concentration episodes were observed in the spring and summer months, particularly July, reflected by the largest seasonal mean and median concentrations occurring in summer ($0.065$ and $0.096\,\mu g\,m^{-3}$, respectively). Mean and median $HNO_3$ concentrations were largest in the spring ($0.16$ and $0.10\,\mu g\,m^{-3}$, respectively), due to a large

peak in mid-April reaching a maximum of $1.45\,\mu g\,m^{-3}$.

### 3.2   Filter radiometer measurements

A direct comparison between the co-located downwelling filter radiometer and $G$ measurements for the duration of the study is presented in Fig. 5. The relationship between the two measurements is linear until $G \approx 500\,W\,m^{-2}$, above which a slight curvature is observed. A quadratic fit to this data yields predicted values of $j(NO_2)$ that lie between two previously published

parameterisations for this relationship. Predicted $j(NO_2)$ values in this study are lower (for $G > 150\,W\,m^{-2}$) than those predicted by the expression of Trebs et al. (2009) shown in Eq. (8), derived from spectral and filter radiometer measurements co-located with $G$ (in $W\,m^{-2}$) at 9 sites worldwide. The $\alpha$ factor is the site-dependent UV-A surface albedo, which here is assumed to be zero as only downwelling radiation is compared. Trebs et al. show differences of up to $50\%$ between their





relationship and Eq. (9), a linear relationship proposed by Bahe et al. (1980) with units of $j(NO_2)$ and $G$ as $min^{-1}$ and

$cal\,cm^{-2}\,min^{-1}$, respectively.

$$j(NO_2) = (1 + \alpha) \times (1.47 \times 10^{-5} \times G - 4.84 \times 10^{-9} \times G^2) \tag{8}$$

$$j(NO_2) = 0.008 + 0.371G \tag{9}$$

While the maximum difference between this study and Trebs et al. is smaller ($\sim 19\%$), predicted values are still larger than those Bahe et al. reported. Trebs et al. (2009) suggest that one possible reason for the lack of curvature observed by Bahe et al.

was that they did not include measurements at SZAs $< 30°$. However Auchencorth Moss is at a higher latitude than any of these sites, with a minimum SZA of $\sim 32°$ in June. Curvature of the plot of $j(NO_2)$ against $G$ is still observed, although there is still a moderate amount of scatter.

The time series of measured and modelled $j(NO_2)$ derived from this study is presented in Fig. 6. In general, model results are larger than the measured counterpart, which is expected as meteorological conditions at Auchencorth Moss are typically

overcast. One clear discrepancy with this generalisation occurs between 30 January and 2 February 2019. This peak in measured data is not reflected in the solar irradiance time series in Fig. 3 as it was largely caused by snow cover at Auchencorth Moss substantially increasing the surface albedo.

The MDAF values calculated as per Eq. (3) from the measured and modelled $j(NO_2)$ was largest during the sunrise and sunset hours, where SZA exceeded $80°$, due to the model predicting very small values of $j(NO_2)$. In general, for the rest of

the year the adjustment factor was greatest in the morning and steadily decreased throughout the day until sunset. The range of values was greatest in the winter months, and the smallest in the summer.

### 3.3    Estimates of $j(HONO)$ and $j(ClNO_2)$

Parameterisations of $j(HONO)$ and $j(ClNO_2)$ based on other more-commonly available variables have been proposed before from both measurement intercomparisons and modelling studies. For example, Kraus and Hofzumahaus (1998) noted that

molecules which photodissociate in a similar spectral region display a higher correlation between rate coefficients. The empirical parameterisation that they derived from spectroradiometer measurements to calculate $j(HONO)$ from measured $j(NO_2)$ (Eq. (10)) has been implemented in further studies requiring $j(HONO)$ estimates (Alicke et al., 2003; Kleffmann et al., 2005; Acker and Möller, 2007).

$$j(HONO) = 0.189 \times j(NO_2) + 8.482 \times 10^{-2} \times [j(NO_2)]^2 \tag{10}$$

A comparison between annual mean diurnal cycles of $j(HONO)$ at Auchencorth Moss parameterised using Eq. (10) and calculated using the TUV model output are presented in the top panel of Fig. 7. Both estimates of $j(HONO)$ have been adjusted using the MDAF metric. Over the entire year, this shows that Eq. (10) is approximately $20\%$ greater than calculated by the TUV model, a result which is consistent over all seasons.

Multiple estimates of $j(ClNO_2)$ have been proposed, including two studies that used parameterisations to update the Master

Chemical Mechanism (MCM; http://mcm.leeds.ac.uk/MCM). Riedel et al. (2014) used measurements made on a research



vessel near California during the CalNex campaign to provide evidence for scaling down $j(NO_2)$ measurements by a factor of 30, while Young et al. (2014) used a linear combination of measured $j(NO_2)$ and $j(O^1D)$, as well as $\sigma(ClNO_2)$ from Ghosh et al. (2012). The MDAF was applied to both of these parameterisations of $j(ClNO_2)$, as well as those developed in this study (TUV model output and updated $j(ClNO_2)$ values; Sect. 2.4.3). The results are shown in Fig. 7.

Overall, the TUV model yields the greatest annual mean $j(ClNO_2)$ values (Fig. 7, bottom panel), exceeding the others by $11-26\%$, with the maximum difference at noon. While the parameterisations from Young et al. (2014) and Riedel et al. (2014) demonstrate reasonable agreement to the updated $j(ClNO_2)$ results, they are closest at different periods during the diurnal cycle. The Riedel et al. parameterisation matches best during peak solar hours (11:00-14:00 UTC), but overestimate by $\sim 15\%$ in the morning (06:00-09:00 UTC) and early evening (17:00-18:00 UTC). In contrast, results from the Young et al. (2014) parameterisation reveal a closer match to updated $j(ClNO_2)$ during the non-peak solar hours, and underestimate by $\sim 5\%$ at noon.

### 3.4 Production rate of OH radicals

Average seasonal production rates of OH radicals ($p(OH)$) from the photodissociations of $O_3$, HONO and $HNO_3$ are presented in Table 3, calculated using $j$-values directly from TUV model output, and with the MDAF applied. In general, the application of the MDAF metric results in $\sim 40\%$ decrease in $p(OH)$ from all considered sources at Auchencorth Moss. In all seasons, $p(OH)_{HNO_3}$ is negligible compared to $p(OH)_{O_3}$ and $p(OH)_{HONO}$, reaching a maximum of about $4.0 \times 10^3$ radicals cm$^{-3}$ s$^{-1}$ in April, coincident with near-maximum concentrations of $HNO_3$ (1.3 µg m$^{-3}$) at the site for the year.

The diurnal variations of these seasonal averages are shown in Fig. 8, where the $\sim 40\%$ reduction due to the MDAF application is clear in daylight hours. The maximum MDAF decrease in hourly measurements is 71%, corresponding to summertime $O_3$ photolysis, resulting in an adjusted $p(OH)_{O_3}$ of $1.90 \times 10^6$ radicals cm$^{-3}$ s$^{-1}$. In comparison, $p(OH)_{HONO}$ in the same month reaches a maximum of $1.03 \times 10^6$ radicals cm$^{-3}$ s$^{-1}$ at 09:00 (UTC; $\sim 5$ hours after sunrise). Conversely, over the daylight hours in winter, $p(OH)_{HONO}$ consistently exceeds $p(OH)_{O_3}$ by a factor of $\sim 5$. A similar pattern is observed in autumn, but at a lower magnitude (factor of $\sim 2.1$). This is likely to be a consequence of the generally overcast conditions typical to Scotland, resulting in the shorter wavelengths of light necessary for $O_3$ photolysis ($\leq 340$ nm) readily scattering before reaching ground-level, *c.f.* longer wavelengths for HONO photolysis ($305 \leq \lambda \leq 420$ nm). This is also observed in spring and summer (Fig. 8), where $p(OH)_{HONO}$ is greater in the early morning ($\sim$04:00-09:00 UTC) and the evening ($\sim$16:00-20:00 UTC). This is is surpassed by $p(OH)_{O_3}$ in the middle of the day, closely following the diurnal cycle of light intensity and peaking when shorter wavelengths are more prevalent. The same diurnal pattern of $p(OH)_{O_3}$ is observed in all seasons (Fig. 9), but a difference of $\sim 25$ is observed between peak $p(OH)_{O_3}$ in summer and winter ($1.90 \times 10^6$ radicals cm$^{-3}$ s$^{-1}$ and $7.4 \times 10^4$ radicals cm$^{-3}$ s$^{-1}$, respectively) despite similar concentrations of $O_3$ (61.0 µg m$^{-3}$ and 57.1 µg m$^{-3}$, respectively).

The photolytic production of OH radicals presented here at Auchencorth Moss is lower than those reported in Melbourne, Australia for March 2017 (Ryan et al., 2018). Compared to the summertime $p(OH)$ rates presented here, peak values of $p(OH)_{HONO}$ were approximately 13 times larger in Australia, while that of $p(OH)_{O_3}$ were just over 50% greater. At both locations, $p(OH)_{O_3}$ peaked at a similar time of day (12:00-14:00 local time), while $p(OH)_{HONO}$ reached a diurnal maximum





much earlier at Auchencorth Moss (09:00 UTC *c.f.* 13:00). The latter is likely due to the application of the MDAF metric, as without the adjustment factor the maximum production rate occurs at noon. Similarly when the MDAF is not applied in calculations of $p(\mathrm{OH})_{\mathrm{O_3}}$, values are comparable at both sites due to equivalent concentrations of $\mathrm{O_3}$ ($\sim 54\,\mathrm{\mu g\,m^{-3}}$ average in Australia). The same is not observed in $p(\mathrm{OH})_{\mathrm{HONO}}$, as $\sim 5$ times larger concentrations of HONO in Australia are a considerable driver towards $\sim 8$ times larger $p(\mathrm{OH})_{\mathrm{HONO}}$ rates.

### 315    3.5    Production rate of Cl atom radicals

Figure 10 presents simulated seasonally-averaged diurnal production rates of Cl radicals, $p(\mathrm{Cl})$, from $\mathrm{ClNO_2}$ photolysis. Plots in the top and bottom panels respectively assume constant concentrations of $\mathrm{ClNO_2}$ ($0.074\,\mathrm{\mu g\,m^{-3}}$), and the diurnally varying $\mathrm{ClNO_2}$ concentration profile shown in Fig. 2. When using the former to estimate $p(\mathrm{Cl})$, those calculated using $j(\mathrm{ClNO_2})$ directly from TUV model output provide the largest estimation for all seasons. When the updated $j(\mathrm{ClNO_2})$ values accounting

for ambient temperature are used (by means of updated $\sigma(\mathrm{ClNO_2})$; Ghosh et al., 2012), total $p(\mathrm{Cl})$ decreased by $\sim 30\%$ compared to the TUV output. As with calculations of $p(\mathrm{OH})$, application of the MDAF metric resulted in $\sim 40\%$ reduction of $p(\mathrm{Cl})$ when using both TUV and updated $j(\mathrm{ClNO_2})$. An exception is in winter, where during the lowest rates of Cl production, the adjustment factor and updated $j(\mathrm{ClNO_2})$ decrease $p(\mathrm{Cl})$ calculated from the TUV output to an approximately equal quantity in the morning hours. This could be due to cooler temperatures in winter ($3.99 \pm 3.6\,^{\circ}\mathrm{C}$) leading to decreased $\sigma(\mathrm{ClNO_2})$ *c.f.*

other seasons ($6.76 \pm 4.3\,^{\circ}\mathrm{C}$, $13.5 \pm 3.7\,^{\circ}\mathrm{C}$ and $7.47 \pm 4.3\,^{\circ}\mathrm{C}$ in spring, summer and autumn, respectively), so that it becomes comparable to TUV results with the MDAF applied.

When the assumed diurnal cycle of $\mathrm{ClNO_2}$ concentrations are used to derive $p(\mathrm{Cl})$ (bottom panels of Fig. 10), the maximum rate of Cl radical production decreases by $59\%$ to $7.98 \times 10^4\,\mathrm{radicals\,cm^{-3}\,s^{-1}}$ compared with when the diurnal cycle of $\mathrm{ClNO_2}$ is unaccounted for. When $\mathrm{ClNO_2}$ concentrations drop close to zero (first occurring between 10:00-11:00 UTC),

there is obviously little photolytic production of Cl. Early morning peaks in $p(\mathrm{Cl})$ in all seasons are staggered, due to the anti-correlated relationship between falling $\mathrm{ClNO_2}$ concentrations and increasing actinic flux with the rising sun at this time of day. In spring and summer evenings from 18:00 onwards, non-zero $p(\mathrm{Cl})$ rates are observed, suggesting that as $\mathrm{ClNO_2}$ begins to form around sunset, the remaining light ($\lambda \leq 475$ nm) quickly results in its photolysis and consequently produces Cl atoms. However, caution should be exercised here as the time of sunset differs between Auchencorth Moss and Weybourne

Atmospheric Observatory by around 15-30 minutes (from the start of spring to the summer solstice). This would impact both the photolysis and formation of $\mathrm{ClNO_2}$, anticipated to lead to $\mathrm{ClNO_2}$ forming later in the diurnal cycle than is estimated in Fig. 2, and thus fewer molecules to undergo photolysis. This discrepancy may mean this apparent late evening $p(\mathrm{Cl})$ is an artefact of the proxy $\mathrm{ClNO_2}$ diurnal cycle.

Similar differences in the magnitude of $p(\mathrm{Cl})$ across the seasons are observed for the $j(\mathrm{ClNO_2})$ and updated values used

in calculation. However, when the MDAF metric is applied, morning hours have a faster rate of Cl production, particularly in spring and summer. Compared to $p(\mathrm{OH})$ with the MDAF applied, the average maximum $p(\mathrm{OH})$ exceed equivalent $p(\mathrm{Cl})$ by $20-25$ times when using $j(\mathrm{ClNO_2})$ output from the TUV model, and $30-35$ times when using the updated $j(\mathrm{ClNO_2})$ values. When considering the diurnal profile of $\mathrm{ClNO_2}$ concentrations, the maximum values of $p(\mathrm{Cl})$ occur earlier in the day ($\sim$06:00



UTC in spring and summer), than $p(\mathrm{OH})_{\mathrm{HONO}}$ (∼09:00-12:00 UTC). This early maximum in $p(\mathrm{Cl})$ is likely driven by the
diurnal cycle of $\mathrm{ClNO_2}$ concentrations in Fig. 2, where concentrations fall after 06:00 UTC. As rate coefficients for reactions
between Cl and VOCs can be up to an order of magnitude larger than comparable rate coefficients with OH, Cl atoms have
potential significant impacts on the radical budget at Auchencorth Moss.

### 3.6 Implications of the MDAF metric

Previous sections describe how application of the MDAF metric at Auchencorth Moss leads to a ∼ 40% reduction in the
$j(\mathrm{HONO})$, $j(\mathrm{ClNO_2})$ and $j(\mathrm{O^1D})$ clear-sky values calculated by the TUV model, and consequently also in the radical produc-
tion rates from these photolyses. In contrast, Sommariva et al. (2020) report measurements of $j(\mathrm{NO_2})$ and $j(\mathrm{O^1D})$ made in
Boulder, Colorado that were $25-30\%$ greater than model estimates using the Master Chemical Mechanism (MCM) parame-
terisations. The difference was attributed to discrepancies between the measurement site and model output location (the latter
is $> 1\,\mathrm{km}$ lower in altitude, and further north by a latitude of $5°$) and to temporal variation in measurements (different seasons
and years). Taken together, however, these authors' findings combined with this study highlight the importance of matched
measurement and model location to generate a MDAF to derive molecule specific $j$-values.

Most local air quality modelling, e.g. for assessment of compliance and mitigation measures, include only simple chemical
schemes with highly parameterised photochemistry (e.g. ADMS). In contrast to model outputted gas and particle concentra-
tions, the values of the model photochemical variables often go unverified. In more detailed instances, atmospheric models
typically use separate radiation models, such as the TUV model used in this study, Fast-JX, or less explicit photolysis schemes
like the Generic Reaction Set (GRS), to approximate $j$-values within the model domain. Radiometer measurements can be used
to validate these model predictions of individual $j$-values. However as these measurements are not typically available, except
as part of specific measurement campaigns, modelled $j$-values are often used unverified.

The implication of the MDAF approach illustrated in this study is that long-term multi-site radiometer measurements could
fill a major gap in measurement and model knowledge. Implementation of such measurements into existing atmospheric trace
gas monitoring networks (e.g. AURN in the UK or EMEP in Europe), would provide invaluable data for understanding tro-
pospheric photochemistry and radical production rates. Accurate $j$-values are integral to accurate assessment of air quality, in
particular, to photolytic production of secondary pollutants which negatively impact both human and environmental health, in-
cluding tropospheric $\mathrm{O_3}$ and particulate matter. Radiometer measurements will consequently have immense value in supporting
atmospheric chemistry measurement and modelling.

### 4 Conclusions

This study presents the first year of a long-term 4-$\pi$ filter radiometer $j(\mathrm{NO_2})$ measurement dataset at Auchencorth Moss, co-
located with relevant ancillary measurements required to determine local radical production rates. Through illustration with the
TUV model, it is demonstrated that long-term filter radiometer measurements can be used (1) to evaluate calculated $j$-values
in current radiation models and parameterisations; and (2) to generate a straightforward measurement-driven adjustment factor



(MDAF) for correcting clear-sky modelled $j$-values for variations in local meteorology (e.g. cloud, aerosol and surface albedo) that would otherwise be both computationally intensive and costly to model.

When the 4-$\pi$ filter radiometer measurement is used to capture these changes in solar flux through the MDAF at Auchencorth Moss, it was found that clear-sky modelled $j$-values at Auchencorth Moss were reduced by $\sim 40\%$ throughout the year.

Quantified production rates of OH radicals in this study differ by approximately one order of magnitude between summer and winter, with the maximum rate in summer reaching $1.90 \times 10^6 \ \mathrm{radicals \ cm^{-3} \ s^{-1}}$ (Auchencorth Moss; $55°$ N). At this rural background site, rate of OH radical production from photolysis of HONO exceeds that from photolysis of $O_3$ during low-light hours, particularly during the sunrise and sunset hours of spring and summer (SZA $> 80°$), and through the full diurnal cycle in winter and autumn (by a factor of 5 and 2.1, respectively). The enhanced contribution from HONO in the colder months is

likely due to Scotland's considerably more overcast conditions in these seasons reducing the shorter wavelengths contributing to $O_3$ photolysis relatively more than the longer wavelengths contributing to HONO photolysis.

Implementing radiometer measurements within existing long-term monitoring networks would provide a higher spatial density of model-measurement $j$-value comparison points in the UK. The consequent more accurate estimations of radical production rates would improve the quantification of subsequent radical-driven chemistry and secondary pollutant generation,

and of all their associated impacts on human health, crops, ecosystems and radiative forcing.

*Data availability.* Filter radiometer measurements for the study year are available in the CEDA Archive (Walker et al., 2020). Concentrations of $O_3$, HONO and $HNO_3$ used were downloaded from UK-AIR (Defra, 2020a). Meteorological measurements used were provided by M. Coyle *et al.* and are available for previous years in the CEDA Archive (Coyle et al., 2020). The TUV model (NCAR, 2019) source code is available for download at: https://www2.acom.ucar.edu/modeling/tuv-download. All figures were created using the *ggplot2* package for R

(Wickham, 2016).

*Author contributions.* HLW, MRH, CFB and MMT devised the study. HLW and IS set up the filter radiometer. SRL, IS, and MRJ collected concentration and other ancillary measurements at Auchencorth Moss. RK ran the spectrophotometer used for calibration. MC provided meteorology data. HLW performed all TUV model runs and analysed the data, with help from MRH, CFB and MMT. The paper was written by HLW with contributions from all co-authors.

*Competing interests.* The authors declare that they have no conflict of interest.

*Acknowledgements.* HLW acknowledges studentship funding from the UK Centre for Ecology & Hydrology (CEH) co-funding of Environment Agency Contract grant NEC05967 and the University of Edinburgh School of Chemistry. The authors acknowledge Dr Carole Helfter for the Phenocam image of Auchencorth Moss, and the UK Department for Environment, Food and Rural Affairs (Defra) and the Devolved





Administrations for maintaining instrument operations (ECM48524). The operation of the $j(NO_2)$ filter radiometer was supported by the UK

Natural Environment Research Council award number NE/R016429/1 as part of the UK-SCAPE programme delivering National Capability.

AURN data from https://uk-air.defra.gov.uk are Crown copyright and licensed under the terms of the Open Government Licence.



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



**Table 1.** Long-term measurement instrumentation at Auchencorth Moss used in this study.

| Variable | Instrument | Reference |
|---|---|---|
| Photolysis rate constant, $j(NO_2)$ | Metcon 4-$\pi$ filter radiometer | (Volz-Thomas et al., 1996) |
| HONO, $HNO_3$ | MARGA[a] | (Twigg et al., 2015; Rumsey et al., 2014) |
| $O_3$ | Teledyne API 400E[b] | (Defra, 2019) |
| Relative humidity (RH) | Rotronics HC2S3 | (Coyle et al., 2019) |
| Temperature ($T$) | Rotronics HC2S3 | (Coyle et al., 2019) |
| Pressure ($p$) | Druck RPT 410F | (Coyle et al., 2019) |
| Global solar irradiance ($G$) | Skye SKS 1110 | (Coyle et al., 2019) |

[a] Monitor for AeRosols and Gases in Ambient Air (Metrohm Applikon, NL).
[b] UV-absorption.


**Table 2.** Summary statistics of hourly trace gas concentrations measured at Auchencorth Moss, for each meteorological season (winter, DJF; spring, MAM; summer, JJA; autumn, SON). Values are mean ($\mu_A$), median ($\mu_M$), minimum, maximum, arithmetic standard deviation ($\sigma_A$) and number of measurements ($n$).

| Trace gas | Season | $\mu_A$ / µg m$^{-3}$ | $\mu_M$ / µg m$^{-3}$ | Min / µg m$^{-3}$ | Max / µg m$^{-3}$ | $\sigma_A$ / µg m$^{-3}$ | $n$ |
|---|---|---|---|---|---|---|---|
| $O_3$ | Winter | 57.1 | 58.6 | 8.2 | 91.4 | 12.5 | 2156 |
| | Spring | 73.9 | 74.2 | 24.0 | 147.8 | 18.7 | 2206 |
| | Summer | 61.0 | 61.3 | 11.6 | 149.7 | 16.3 | 2205 |
| | Autumn | 50.3 | 51.2 | 2.2 | 76.2 | 12.2 | 1963 |
| HONO | Winter | 0.076 | 0.061 | 0.020 | 0.54 | 0.053 | 1567 |
| | Spring | 0.076 | 0.059 | 0.003 | 0.53 | 0.058 | 1916 |
| | Summer | 0.091 | 0.065 | 0.016 | 0.57 | 0.074 | 2036 |
| | Autumn | 0.069 | 0.059 | 0.007 | 0.70 | 0.045 | 1769 |
| $HNO_3$ | Winter | 0.096 | 0.056 | 0.001 | 1.55 | 0.14 | 1567 |
| | Spring | 0.16 | 0.10 | 0.003 | 1.46 | 0.20 | 1916 |
| | Summer | 0.10 | 0.084 | 0.001 | 0.67 | 0.081 | 2036 |
| | Autumn | 0.067 | 0.052 | 0.001 | 0.56 | 0.062 | 1769 |


**Table 3.** Mean production rates of OH from the photolysis of $O_3$, HONO and $HNO_3$ during each meteorological season, using $j$-values directly output from the TUV model and those adjusted using the MDAF metric. Means are calculated using only daytime measurements, and are reported with corresponding confidence intervals (95%). The % decrement in production rate for the MDAF data is the average decrement across the season.

| Season | $p(\mathbf{OH})_{\mathbf{O_3}}$ / $\times 10^4$ radicals cm$^{-3}$ s$^{-1}$ | | | $p(\mathbf{OH})_{\mathbf{HONO}}$ / $\times 10^4$ radicals cm$^{-3}$ s$^{-1}$ | | | $p(\mathbf{OH})_{\mathbf{HNO_3}}$ / radicals cm$^{-3}$ s$^{-1}$ | | |
|---|---|---|---|---|---|---|---|---|---|
| | **TUV** | **MDAF** | $\Delta$ / % | **TUV** | **MDAF** | $\Delta$ / % | **TUV** | **MDAF** | $\Delta$ / % |
| Winter (DJF) | $5.08 \pm 0.55$ | $3.16 \pm 0.35$ | 24 | $24.3 \pm 2.5$ | $18.1 \pm 2.2$ | 24 | $27.3 \pm 4.7$ | $19.9 \pm 3.6$ | 24 |
| Spring (MAM) | $68.2 \pm 4.4$ | $37.1 \pm 2.8$ | 41 | $69.4 \pm 4.5$ | $41.1 \pm 3.2$ | 40 | $321 \pm 34$ | $204 \pm 26$ | 40 |
| Summer (JJA) | $110 \pm 6.1$ | $60.0 \pm 3.7$ | 46 | $78.1 \pm 4.6$ | $46.8 \pm 3.0$ | 47 | $214 \pm 15$ | $122 \pm 10$ | 47 |
| Autumn (SON) | $24.6 \pm 2.1$ | $13.1 \pm 1.3$ | 41 | $42.6 \pm 4.1$ | $22.5 \pm 1.8$ | 40 | $47.4 \pm 5.4$ | $27.5 \pm 4.0$ | 40 |



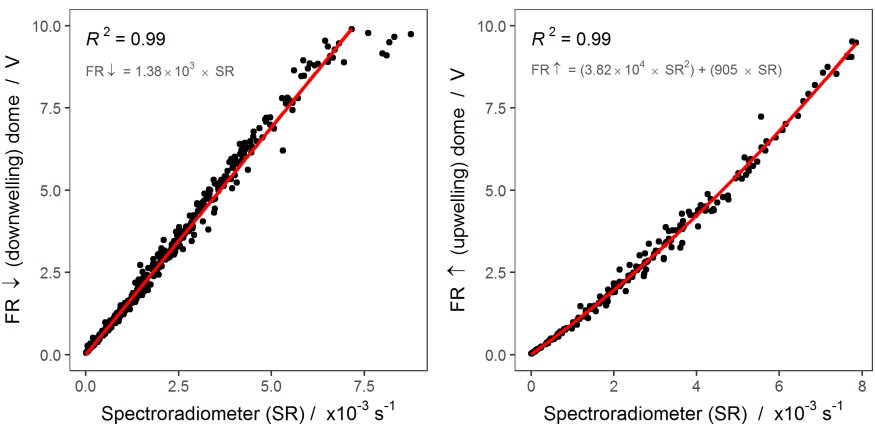

**Figure 1.** Calibration of both filter radiometer optical inlet domes against a Bentham DTM300 spectroradiometer from 13–25 June 2019. Filter radiometer measurements (1 s) are averaged to equal the scan duration of spectroradiometer (approx. 3 mins). Relationships used for subsequent conversion of filter radiometer measurements are presented in red.



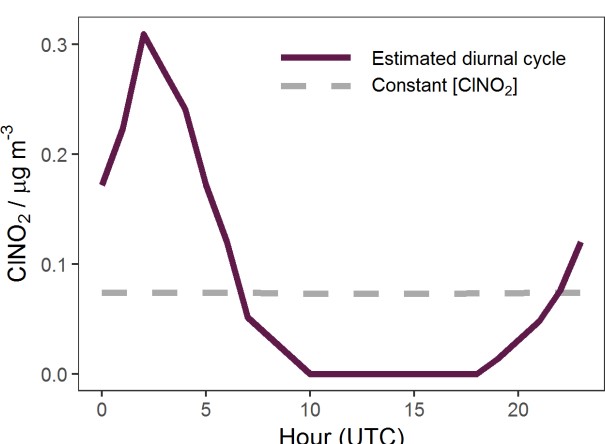

**Figure 2.** Diurnal cycle of $ClNO_2$ concentration estimates used to calculate the production rate of Cl radicals from $ClNO_2$ photolysis. Diurnal cycle based on summer Weybourne measurements from Sommariva et al. (2018).



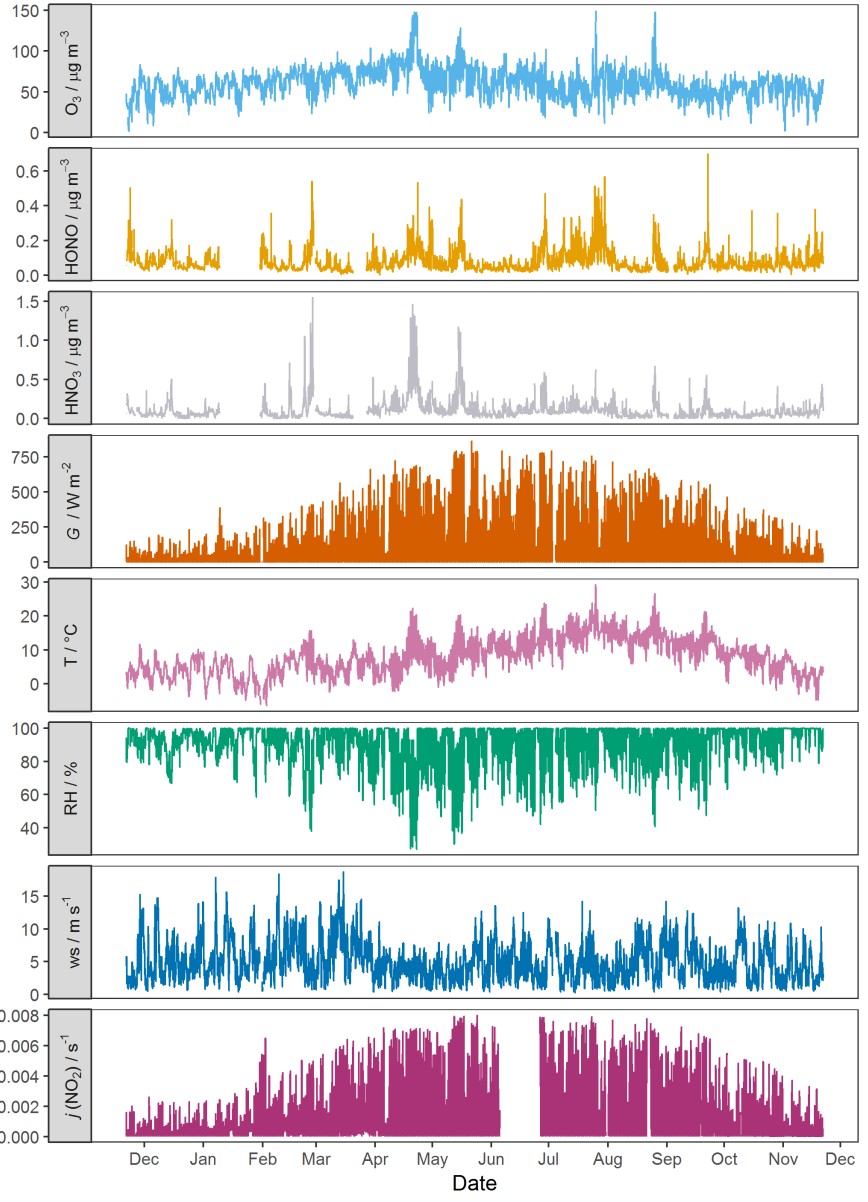

**Figure 3.** Time series of mean hourly trace gas concentrations ($O_3$, HONO, $HNO_3$) and meteorological parameters (solar global irradiance ($G$), air temperature, relative humidity and wind speed) for the year of data included in this study. The $j(NO_2)$ measurements reported are the hourly mean of summed down- and up-welling components.



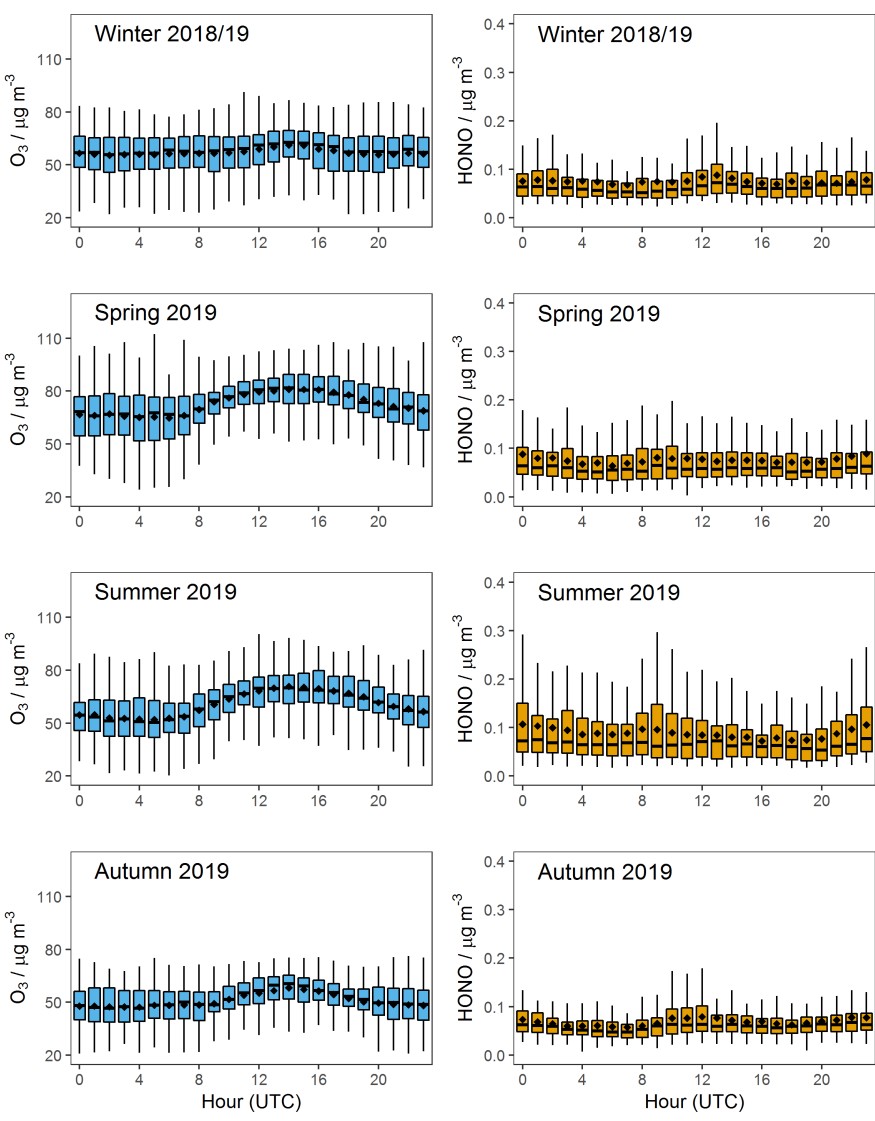

**Figure 4.** Median hourly diurnal trace gas concentrations of $O_3$ and HONO in each meteorological season. Mean values are represented by diamond points. Boxes show the upper and lower quartiles, and the whiskers present the 5-95% range. Time in all seasons refer to UTC.

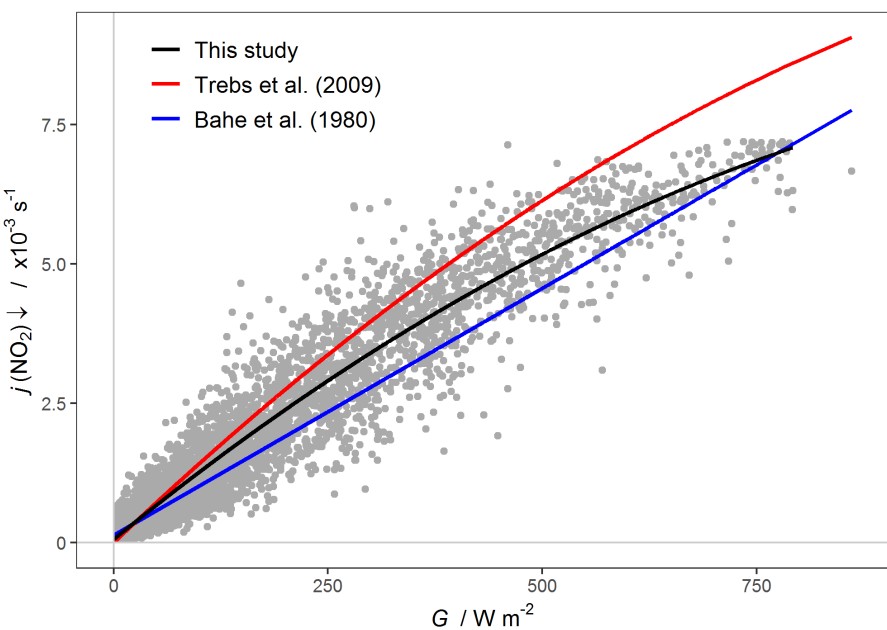

**Figure 5.** Scatter plot of hourly averaged down-welling $j(NO_2)$ measured by the upward facing dome of the filter radiometer, and the solar global irradiance ($G$) for the year of data collected. The second-order polynomial fit to these data is shown in black. The parameterisations between $j(NO_2)$ and $G$ from Bahe et al. (1980) and Trebs et al. (2009) are also shown for comparison.

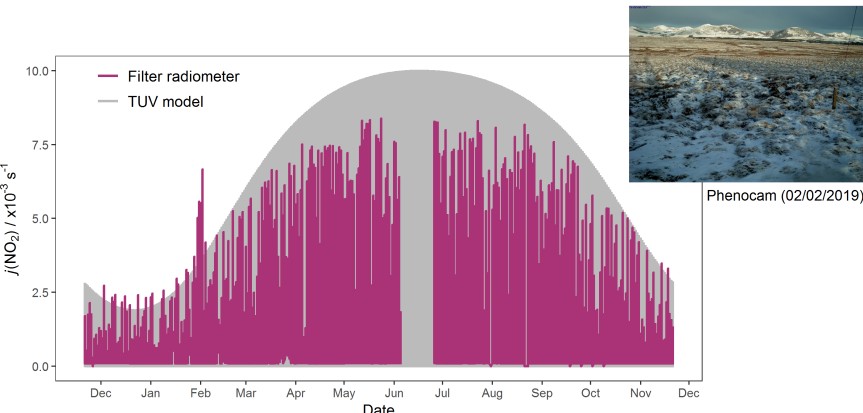

**Figure 6.** Comparison of hourly TUV modelled $j(NO_2)$ at Auchencorth Moss, and total down- and up-welling $j(NO_2)$ as measured by the filter radiometer at Auchencorth Moss for the year of study. Inset shows a photo of the site south of the filter radiometer taken on 02/02/2019 by an automatic Phenocam (EuroPhen, 2020)

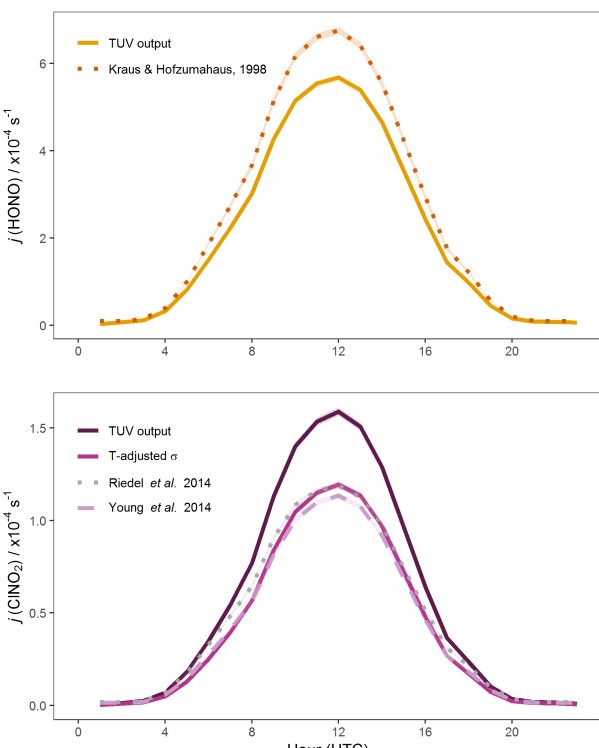

**Figure 7. Top:** Annual average diurnal cycle of $j$(HONO) at Auchencorth Moss calculated by the TUV model and using the Kraus and Hofzumahaus (1998) parameterisation. **Bottom:** Annual average diurnal cycle of $j$(ClNO$_2$) estimates for Auchencorth Moss, using the direct TUV model output and updated $j$(ClNO$_2$) values (accounting for daily average temperature), and parameterisations presented in Riedel et al. (2014) and Young et al. (2014). All values of $j$(HONO) and $j$(ClNO$_2$) are the annual mean of each hour with the MDAF metric applied, to account for local influence on solar fluxes. 95% confidence intervals shaded.



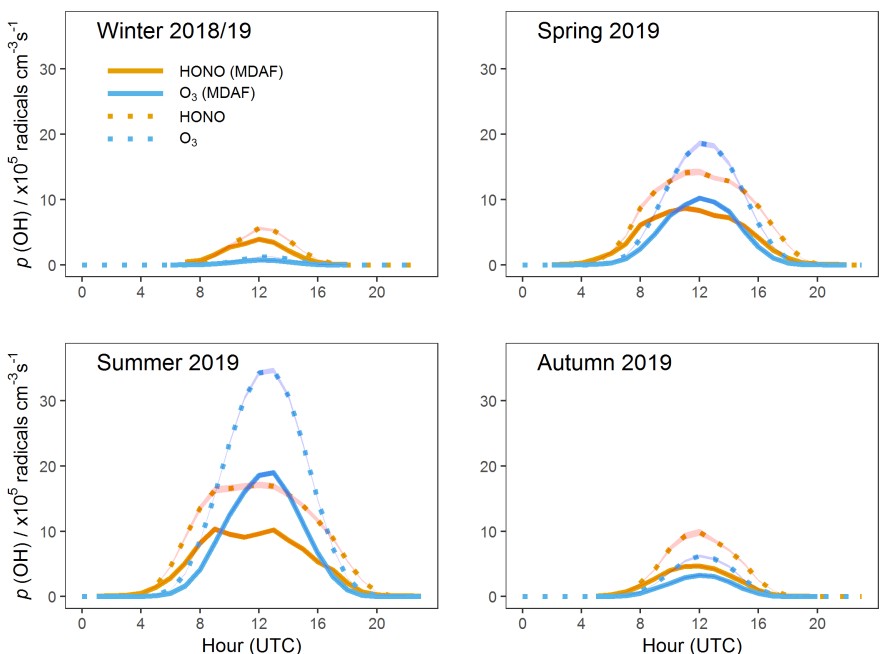

**Figure 8.** Diurnal variations in seasonally-averaged hourly mean $p(\mathrm{OH})$ rates from $O_3$ and HONO photolysis. Dotted lines show rates calculated using $j$-values directly from the TUV model, while solid lines show rates where $j$-values were first corrected by the MDAF metric, as described in the text. Shading represents the 95% confidence intervals.





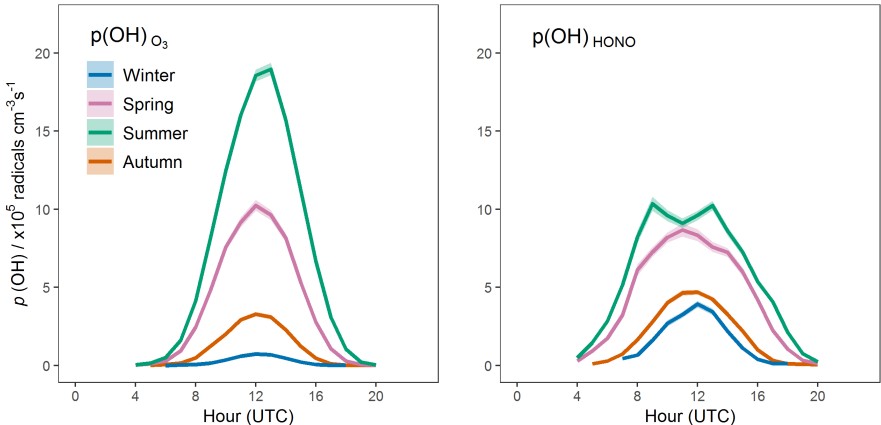

**Figure 9.** Diurnal variations in seasonally-averaged hourly mean $p(\mathrm{OH})$ rates from $O_3$ and HONO photolysis, where $j$-values used are corrected by the MDAF. Shading represents 95% confidence interval.



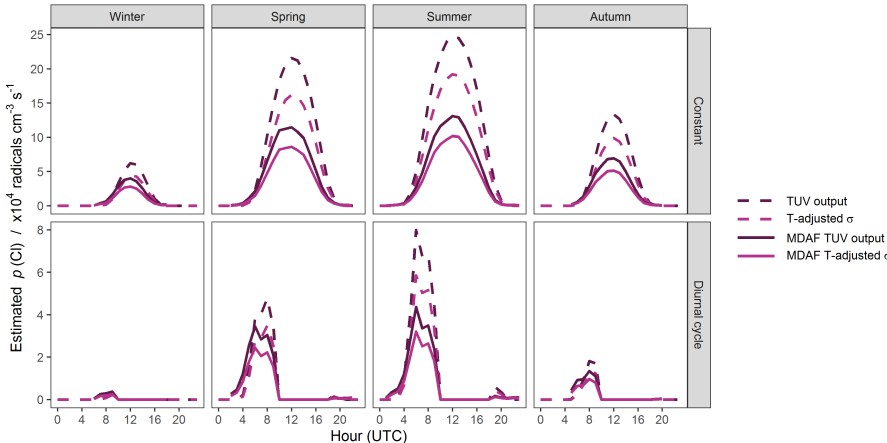

**Figure 10.** Diurnal variations in seasonally-averaged hourly production of chlorine atom radicals, $p(\text{Cl})$, from nitryl chloride, using a constant $\text{ClNO}_2$ concentration of 21.5 ppt (upper panel), and the diurnal cycle of $\text{ClNO}_2$ presented in Fig. 2 (lower panel).