# Peer review of "Use of filter radiometer measurements to derive local photolysis rates and for future monitoring network application"

_Atmospheric Measurement Techniques, 2020_

## Referee Comment (RC1) · Anonymous Referee #1 · 8 Jul 2020

General comments

The paper describes the use of filter radiometer measurements of j(NO2) to derive photolysis rate coefficients and primary production rates of OH radicals (and Cl atoms) from various precursors at a site in Scotland for a period of one year. The study additionally uses clear sky TUV model calculations of actinic flux, local measurements of precursors and meteorological data including solar shortwave irradiance. The general idea is to utilize measured and clear sky modelled values of j(NO2) to calculate adjustment factors (MDAF), suitable to derive local photolysis rate coefficients for species other than NO2. These rate coefficients were then applied to determine radical pro-

duction rates for some measured (and unmeasured) precursors. The MDAF approach is in principle sound. However, the study in my view has major deficiencies and should not be published in the present form. My main objections are listed below followed by some minor and technical comments.

Section 2.2: Filter radiometer

As is noted in the text (line 142 ff) and visible in Fig. 1, the filter radiometer instrument covering downwelling radiation reached signal voltages above 10 V that were not properly recorded, presumably because the range of the data logger was limited to 10 V. Even though this was noticed (at the latest) during the calibration in June, the problem was apparently not fixed. Basically this means that $j(NO2) > 7E-3/s$ were not recorded (Fig. 1) which is well below expected clear sky maximum values of around $8E-3/s$ for the downwelling in summer. Moreover, under conditions of broken clouds there is the effect of short-term enhancements that can be observed for solar shortwave irradiance (e.g. Schade et al., 2007) but also for $j(NO2)$. This probably led to occasional short-term values greater than $7E-3/s$ even at lower sun elevations. Looking at one hour averages alone the effect may remain unnoticed but the data are nevertheless biased. So the problem is not limited to three weeks in June (that were probably for that reason taken out later in Figs. 3 and 6 without explanation). I think the authors have to go back to their original 1 s data set of FR and flag all data where within the one hour averaging periods 10 V values occurred. Consequently these periods have to be removed from the further data analysis and mean diurnal variations for spring and the summer periods may become disputable. Moreover, this will affect the correlation with the solar shortwave irradiance in Fig. 5 (see below).

Section 2.3: Model approach

The use of an annual mean value of ozone columns as a TUV model input (line 162) is an unnecessary simplification. The ozone column is a major factor influencing $j(O1D)$ and $j(HNO3)$ which is not covered by the $j(NO2)$ measurements and is therefore not included in the MDAFs. Ozone columns are highly variable, especially during spring time in Europe and are apparently available for the site on a daily basis (or from satellite observations). Modelled j(O1D) should also take into account ambient temperature which mainly affects the quantum yields of O(1D). The temperature effect was considered for ClNO2 (mentioned later in line 206). It's unclear why it was neglected for the more important j(O1D). It is also not explained why MDAFs were derived in the first place for an unimportant precursor like HNO3 and an unmeasured precursor like ClNO2 but not for a species like HCHO, a well-known and important primary radical source in this case of HO2.

Section 2.4.1: Adjustment factor

The quality of the applied adjustment factors (MDAF) remains unclear and is not investigated even though it is a key parameter of this work. For example, the calculated clear sky-photolysis rate coefficients j(NO2) and j(O1D) will generally not correlate well because of the different spectral regions where the photolysis takes place which results in different dependencies on solar elevation. In addition j(O1D) is affected by temperature and ozone columns while j(NO2) is not (or very little). The uncertainty of the MDAF should be carefully estimated for each photolysis rate coefficient dependent on solar elevation and atmospheric conditions including cloud effects. Without such sensitivity studies the uncertainties of the adjusted j-values remain unclear, as well as the benefits of the whole approach.

Section 2.4.3 Production rate of Cl atoms

ClNO2 was not measured at the site (line 211). There is no reason to assume that ClNO2 concentrations are similar to those measured in Weybourne unless at least NOx and O3 levels are comparable (not shown), let alone to consider a constant mean concentration or a seasonably independent mean diurnal concentration. Consequently, the results presented later in Sect. 3.5 are pure speculation. j(ClNO2) should of course be considered as an important photolysis rate coefficient but the attempt to apply it for

the site without ClNO2 measurements is pointless (see below).

Section 3.2 Filter radiometer measurements

Because of the limitations of the measurement technique as discussed above (Sect. 2.2), the current j(NO2) dataset does not seem suitable to derive an empirical relationship between j(NO2) and solar shortwave irradiance. Fig. 5 clearly shows that data points above the instrument's limit are missing. Moreover, as noted, some of the 1h averages shown may be biased low. So the agreement with the parametrization by Trebs et al. may in fact be better than implied by the blue line. The authors seem to recognize that because they reproduce the empirical formulas from previous studies in the text but not their own. The employed pyranometer also requires more attention. The SKS 1110 is not a standard pyranometer. It has a fairly limited spectral response range much more narrow than the solar range and is probably calibrated by the manufacturer by comparison with a reference pyranometer in the field. According to Coyle et al., 2019, the instrument was running for more than 20 years at the site without a follow-up calibration. Such a calibration was suggested by Coyle et al., 2019. Was it made and implemented in the current work?

Section 3.3 Estimates of j(HONO) and j(ClNO2)

The discussion of the empirical relationship between j(NO2) and j(HONO) by Kraus et al. is misleading. The fact that this older parametrization gives higher values than the TUV model is not a result related with the j(NO2) measurements of the present study. It's just a property of the TUV model which used more recent recommendations of absorption cross sections and quantum yields. Qualitatively the discrepancy is explainable because Kraus et al. used recommendations from 1998: smaller quantum yields for the NO2 photolysis and greater absorption cross sections for HONO. What is more important for this study is that j(NO2) and j(HONO) from TUV probably correlate linearly in good approximation which justifies the use of the MDAF concept for j(HONO).

For j(ClNO2) a similar problem arises: The result that TUV overestimates j(ClNO2) has again nothing to do with the j(NO2) measurements. It's simply because the implemented absorption cross sections are obviously outdated as noted in Sect. 2.3 of the paper. How much greater the values are is secondary for the current study but it could of course be mentioned in the text as a friendly reminder for the TUV developers to update their input.

A discussion of j(O1D) is completely missing here. The points raised in Sect. 2.3 and 2.4.1 should be considered and a reasonable uncertainty estimate should be derived for all photolysis rate coefficients. The MDAF concept is far more complicated and error-prone in particular for j(O1D) than currently implied in the paper.

Section 3.4 and 3.5: Production rate of OH radicals / Production rate of Cl atoms

That HNO3 photolysis is insignificant (line 290) as an OH precursor is by no means new information and could have been estimated from clear sky summer maximum OH production rates alone. The impression is that HNO3 was included in this study only because it was measured.

The explanations in lines 297-305 regarding the different seasonal dependencies of p(OH) from O3 and HONO are not reproducible. Cloud effects may be a factor but the main seasonal difference is that j(O1D) and j(HONO) depend differently on solar elevation. The diurnal variations of p(OH) from O3 are probably not the same in summer and winter (line 303). The authors will notice that when they multiply the winter data with 25 (line 304).

The comparison of the obtained p(OH) with those from a single study in Australia is arbitrary. That HONO photolysis is a significant or even dominant OH precursor (at ground level) is well known and was discussed in many well designed studies before. The current paper gives little new insight as the quality of the HONO measurements remains unclear as well as local NOx levels that could classify the results.

As was noted already in Section 2.4.3 the discussion of Cl production rates based on devised diurnal concentrations is too speculative. So the lengthy discussion on possible diurnal or seasonal variations of Cl production at the site is not productive. That ClNO2 photolysis can lead to an increased VOC oxidation in particular during morning hours has been discussed in a number of previous studies, however based on ClNO2 measurements. ClNO2 concentrations are typically extremely variable. Compared to that potential uncertainties related with photolysis rate coefficients seem to be a minor problem. The final sentence in line 345 intended to emphasize the (potential) impact of Cl atoms for the radical budget at the site is also misleading. The fact that Cl atoms react faster with many VOCs than OH is secondary. Ultimately they both react mostly with VOCs and what counts are the production rates.

Overall the sections on the radical production rates are not very conclusive. The production rates are important parameters but they are just a small fraction of production and destruction rates during daytime that make up tropospheric photochemistry. So the absolute numbers are difficult to interpret unless you run a box model for comparison or you manage to extract these parameters from an operational model for this location. Moreover, this work is not about the concentrations of the precursors and whether or not they are represented correctly in current models. If you would just discuss the photolysis rate coefficients, a very similar picture would emerge regarding the importance of local effects on radiation. In accordance with that, the whole introduction is concerned with radiation, not with radical production rates.

Section 3.6 and 3.7: Implications of the MDAF metric / Conclusions

The paragraph following line 351, "In contrast, Sommariva et al. report…." is confusing: MCM parametrizations, different altitudes, and different seasons all mixed together. I don't think this arbitrary citation is necessary to emphasize the need for locally measured photolysis rate coefficients for a proper analysis of field data.

The paragraphs line 364-370 and 387-390 where "long-term multi-site radiometer mea-

surements" (line 364) are suggested, apparently draft a main conclusion of the study which is also implied in the title. However, I don't think such an investment would be justified based on the current analysis. The authors themselves show that a fairly compact empirical relationship exists between j(NO2) and solar shortwave radiation (Trebs et al., 2009). Even though this parameterization is not perfect, it will nevertheless cover a large fraction of the most important local cloud effects. As the authors also state, pyranometers are already used within existing networks and have a high degree of standardization. So to "provide a higher spatial density of model-measurement j-value comparison points in the UK" (line 387) a first step could be to use these already available long term data from many stations to evaluate the model data and to investigate the impact of potential shortcomings of model-predicted j-values. If it then turns out that the j(NO2) derived from the pyranometer data are not accurate enough for this purpose you would have a point. But I doubt this will be the case.

Specific comments

Line 5: "In this paper, locally representative photolysis rate constants (j-values) for these molecules are shown to be critical for quantifying and understanding the rate of radical production in a local atmosphere." Is this a new insight? j-values are routinely measured since decades during photochemical campaigns for exactly that reason.

Line 55: "Cl radicals are extremely reactive..., Cl has a significant effect on local tropospheric oxidation." I think this statement is misleading. Low Cl concentrations may be a result of the high reactivity of Cl but the importance of Cl depends on its production rate. That it's more reactive than OH is unimportant (moreover, OH is regenerated which makes it the key oxidant).

Line 102: "...when measured AOD and NO2 concentrations are included in model inputs, actinic flux is reasonably well predicted..." It should be clarified that this statement probably refers to clear-sky conditions.

Line 134: Give a reference where the Bentham instrument and its uncertainties are

described or provide more information.

Line 170: Why is Section 2.4 entitled "Theoretical calculations"? In this Section you combine measured data with those from the TUV model.

Line 223: "UK seasons are defined. . ." Is this definition useful with regard to radiation measurements?

Line 247, Eq. (9): Convert to units used in this work.

Line 258: "The MDAF values . . . was largest during sunrise and sunset hours" This is explainable by an instrumental artefact. At low sun direct radiation can strike the upper and the lower dome simultaneously unless your "artificial horizon" is large enough to prevent this. For the common Metcon instruments it is not.

Line 260: ".. the adjustment factor was greatest in the morning and steadily decreases throughout the day until sunset". Is there an explanation for this behaviour? The mean diurnal variations of j(HONO) and j(ClNO2) in Fig. 7 don't show this.

Line 294: I do not see the 71% decrease in Fig. 8. It looks more like the overall 40% decrease stated in line 293.

Line 385ff: "The enhanced contribution. . ." Please reconsider this statement taking into account the different dependencies of j(O1D) and j(HONO) on solar elevation. The clear-sky TUV results will show the same effect.

Figure 3: The gap in the j(NO2) data should be noted in the caption.

Figure 6: What can be recognized in this figure are daily maximum values rather than hourly data. A correlation would perhaps be more insightful. The gap in the data requires an explanation.

Figure 7, 8, 9: The 95% confidence intervals are virtually invisible and it's unclear what they represent.

References

N. H. Schade, A. Macke, H. Sandmann and C. Stick, Enhanced solar global irradiance during cloudy sky Conditions, Meteorologische Zeitschrift, Vol. 16, No. 3, 295-303 (June 2007)
* * *

---

## Referee Comment (RC2) · Anonymous Referee #2 · 17 Jul 2020

General Comments: The work ratios long-term filter radiometer jNO2 measurements to a cloud-free radiative transfer model (TUV) to calculate a measurement-driven adjustment factor (MDAF). Modeled jO3, jHONO, jClNO2 and jHNO3 are then multiplied (presumably) by the MDAF correction factor to determine the environmentally impacted local values. Combining these rates with local measurements, the authors calculate radical products and find a significant reduction in OH compared with the cloud-free modeled products. The authors suggest that such local radiation measurements would significantly improve chemical model calculations of important species.

The local impact of clouds, aerosols and changing albedo on photolysis rates and

the resulting impacts on radical chemistry require additional validation in models. The long-term dataset discussed provides the opportunity to examine a range of conditions to test model sensitivity to changes in local conditions and the incentive to set up additional jNO2 measurement locations. The paper is relevant but lacks rigor in some areas. With revisions and additional discussion the paper would be worthy of publication.

Specific comments: Line 13: 40% lower OH production rate compared to what? Presumably compared to the cloud-free model. Best for clarity to be explicit. Some discussion of cloud-free vs cloud resolving models would be useful.

Line 35: I suggest an alternative PAN reference:

Singh, H. B., Herlth, D., O'Hara, D., Zahnle, K., Bradshaw, J. D., Sandholm, S. T., Talbot, R., Crutzen, P. J., and Kanakidou, M.: Relationship of Peroxyacetyl nitrate to active and total odd nitrogen at northern high latitudes: Influence of reservoir species on NOx and O3, J. Geophys. Res, 97, 16523–16530, 1992.

Line 70: Many molecules include a pressure dependence. I suggest adding pressure (p) to equations.

Line 72: The definition attributed to Madronich is confusing. Actinic flux is more simply the spherically integrated radiation through a sphere or the radiant energy density incident on a unit spherical surface.

Line 78: Here and throughout the paper, the authors refer to applying or using the MDAF. However, I don't see the "application" is ever explicitly defined. I am left to presume (perhaps obviously) that MDAF is a simple multiplication factor. That should be stated here.

Line 86: This seems like an odd line noting that jNO2 instruments have "limited potential to estimate the photolysis frequencies of other atmospheric species." Isn't that the point of this paper? I think you are saying that they cannot *directly* measure the other

species without MDAF or a similar correction.

Line 103: This is an incorrect oversimplification. The Palancar data was screened for clouds and included PBL heights in addition to AOD and NO2 concentrations.

Line 127: The difficulties of a 4 pi spectrometer should be discussed. In particular, the upwelling can be significantly influenced by the support tower, any nearby equipment and the local albedo (e.g. a bush or rock below the tower). This can add significant uncertainty, particularly when the upwelling is large (e.g. snow). Also, the errors in the two 2-pi optics are particularly large near the horizion. This is noted throughout the literature (e.g. Hofzumahaus, et al, (2002) doi:10.1029/2001JD900142 and references therein). See the comment at lines 258-61.

Lines 146-9: I recommend being explicit about the averaging for clarity. The filter radiometer is broadband and continuously measures the full jNO2 spectrum. The scanning spectrometer measures one narrow wavelength band linearly in time over the 3 minute duty cycle. Thus, the measurements are not equivalent in rapidly changing conditions.

Line 158: Be consistent TUV 5.3 or 5.3.1 (as noted in the abstract).

Line 161: The model is not technically "clear-sky" because aerosols are included. I suggest using "cloud-free" throughout.

Lines 163-4: List the default values for albedo and aerosols (presumably 0.10 and the Elterman (1967) continental aerosol profile).

L. Elterman UV, Visible and IR Attenuation to 50 Km, AFCRL-68-0153, Environ. Res. Papers, (No. 278) (1968), Bedford, Mass.

Lines 168-9 : I believe default TUV uses a different cross section (Vandaele et al., 1998). According to JPL 2015, Vandaele and Mérienne are within 2-3% across their measurement spectral range.

[Figure]

Line 190: Describe kH2O, kN2 and kO2.

Line 207: Eq 2 is specific to jNO2. One option would be to make Eq 2 generic and adjust the ("in this case") text in line 75.

Line 215: Discuss expected seasonal variation in ClNO2 as Sommariva was summertime only.

Line 258-61: A figure would be helpful to understand the SZA relationship to MDAF. At low sun, the angular response of the radiometer optics will contribute significantly to the MDAF. How does the filter radiometer data compare to the Bentham at low sun? In addition, the model will be particularly sensitive to the aerosols applied in TUV (Elterman). These will generally differ from the actual profile (including the PBL height). I expect the MDAF analysis is not particularly effective at very low sun (perhaps >85 deg sza). This topic should be addressed and perhaps such data needs to be excluded.

Line 287: The discussion of HNO3 is a bit odd. Either it should be given the full analysis of the other molecules or it should excluded because it is not relevant to the OH analysis. The low OH production from HNO3 is not even mentioned in the conclusion, perhaps because it is not an interesting finding of the study. If HNO3 is included, why not a list of other photolysis frequencies?

Line 289 and Table 3: I think this must state "cloud-free TUV" for clarity.

Line 295: jNO2 and jO3 cover significantly different spectral ranges resulting in differing diurnal profiles and interactions with clouds and aerosols. More importantly, jO3 is highly dependent on the O3 column. Applying the MDAF to jO3 will result in significant uncertainty and I question whether it is a valid method. For more, see Lefer et al., 2003 (doi:10.1029/2002JD003171). The jClNO2 and jHONO have spectral parameters similar enough to jNO2 that MDAF is likely ok. The paper lacks any discussion of uncertainties resulting from the MDAF results for each of the molecules.

Lines 364-70: This would be a good place to discuss using MDAF for an expanded set

of photolysis frequencies beyond those listed in this paper. The metric would be useful for other studies.

Fig 6: What happened in June? Does this affect the annual averages shown in figure 7?

Technical corrections: Line 76: Change "and temperature" to "temperature (T) and pressure (p)

Line 144: Shouldn't this be ">9V"

Line 166: Perhaps, note that is is below, "shown in Eq. (3) (below in Section 2.4.1)"

Lines 174-5: Remove "it has been demonstrated that" and change "could" to "can"

Line 258: Change "was" to "were"

Line 316 and Fig 10: Previously used only "p(Cl)ClNO2" but now shortened to "p(Cl)". I suggest sticking to the long version.

Line 328-9: Reword for clarity "compared with when the diurnal cycle of ClNO2 is unaccounted for" to "compared with the constant ClNO2 concentration"

Fig 8: Legend should say "HONO (cloud-free TUV)" and "O3 (cloud-free TUV)" for the dotted values

[Figure]

---

## Author Comment (AC1) · 24 Sep 2020

**amt-2020-219: Use of filter radiometer measurements to derive local photolysis rates and for future monitoring network application (Walker *et al.*)**

**Responses to Reviewer 1**

The authors thank Reviewer 1 for their time spent conducting a careful review of our paper, and providing constructive comments. We hope that our comments and corresponding revisions to the original manuscript are satisfactory.

The major revisions to the manuscript after comments from both Reviewer 1 and 2 include the following.

- Re-running of the TUV model to include measurements of daily average ozone column and ambient temperature (see our response to comments on Section 2.3).
- Estimates of $p(Cl)_{ClNO_2}$ rates have been excluded and Section 3.5 (*"Production rate of Cl atom radials"*, submitted manuscript) has been removed (see response to Section 2.4.3 and 3.4/3.5).
- A new section has been introduced (*"Section 3.3: MDAF derivations"*, revised manuscript) to detail methods of determining the uncertainty arising from applying a $j$(NO₂)-derived MDAF, as we present here, to other species $j$-values (see response to Section 2.4.1).

Our responses to individual points raised by the review are listed below. The suggestions and comments made by the reviewer are listed in black font, and our responses highlighted in blue, with any relevant manuscript changes indicated in *blue italic*. We have also amended the sub-sectional structuring, and some other text (e.g. abstract) of our manuscript as a consequence the revisions we describe above and below.

**General comments**

The paper describes the use of filter radiometer measurements of j(NO2) to derive photolysis rate coefficients and primary production rates of OH radicals (and Cl atoms) from various precursors at a site in Scotland for a period of one year. The study additionally uses clear sky TUV model calculations of actinic flux, local measurements of precursors and meteorological data including solar shortwave irradiance. The general idea is to utilize measured and clear sky modelled values of j(NO2) to calculate adjustment factors (MDAF), suitable to derive local photolysis rate coefficients for species other than NO2. These rate coefficients were then applied to determine radical production rates for some measured (and unmeasured) precursors. The MDAF approach is in principle sound. However, the study in my view has major deficiencies and should not be published in the present form. My main objections are listed below followed by some minor and technical comments.

We are pleased the reviewer thinks that the MDAF approach is sound. We highlight that the paper also presents our work in the context of discussing the basis for the use of in-situ photolysis measurements for long-term validation and local atmospheric chemistry studies. We have carefully considered the reviewer's comments on deficiencies and objections and taken forward many of the reviewer's suggestions. We think that the adjustments have significantly improved the manuscript and addressed the scientific issues raised.

To improve clarification of the aims of our study, we have revised the aims at the end of the introduction in the revised manuscript.

*Section 1: "This study presents the first long-term j(NO₂) filter radiometer dataset in the UK (Auchencorth Moss, SE Scotland), its use in generating a measurement-driven*

*adjustment factor (MDAF) and how this can be applied to basic model estimations of j-values at a local site. The filter radiometer dataset is shown to account for variations in local meteorological conditions which impact j(NO₂) (e.g. cloud cover, surface albedo), and how this information within the MDAF can be used as a point of model validation or education, with a minimal increase in computational cost. The impact of the MDAF on atmospheric chemistry is discussed in the context of unimolecular photolysis reactions, and hydroxyl (OH) radical production rates from precursor sources routinely measured at Auchencorth Moss (O₃, HONO and HNO₃). The scientific, and ultimately policy, value of incorporating filter radiometer measurements alongside existing long-term and spatially-resolved air quality monitoring networks is emphasised.”*

**Major comments**

*Section 2.2: Filter radiometer:* As is noted in the text (line 142 ff) and visible in Fig. 1, the filter radiometer instrument covering downwelling radiation reached signal voltages above 10 V that were not properly recorded, presumably because the range of the data logger was limited to 10 V. Even though this was noticed (at the latest) during the calibration in June, the problem was apparently not fixed. Basically this means that j(NO2) > 7E-3/s were not recorded (Fig. 1) which is well below expected clear sky maximum values of around 8E-3/s for the downwelling in summer. Moreover, under conditions of broken clouds there is the effect of short-term enhancements that can be observed for solar shortwave irradiance (e.g. Schade et al., 2007) but also for j(NO2). This probably led to occasional shortterm values greater than 7E-3/s even at lower sun elevations. Looking at one hour averages alone the effect may remain unnoticed but the data are nevertheless biased. So the problem is not limited to three weeks in June (that were probably for that reason taken out later in Figs. 3 and 6 without explanation). I think the authors have to go back to their original 1 s data set of FR and flag all data where within the one hour averaging periods 10 V values occurred. Consequently these periods have to be removed from the further data analysis and mean diurnal variations for spring and the summer periods may become disputable. Moreover, this will affect the correlation with the solar shortwave irradiance in Fig. 5 (see below).

We thank the reviewer for the detailed assessment, and we address the specific issues raised individually.

**1) Handling and validation of 1 second data**

We remind the reviewer that in calculating hourly average values of measured $j(NO_2)$, the original 1 second dataset was flagged as invalid where measurements in the downwelling dome exceeded the topping out threshold. This comprised 1.4% of the full dataset (line 147, submitted manuscript). For information, the distribution of invalid flagged data is shown in Figure AC1 below.

In our revised manuscript, an hourly average has *only* been quantified if the 1 second data collected within that hour met or exceeded 75% data capture, after all invalid flagged data has been excluded. The manuscript has been updated to detail the data handling procedures used.

*Section 2.2: "These incidences of high j(NO₂)↓ (>9 V) comprised 1.4% of all raw data collected, and were excluded from further analysis. An hourly average was only calculated if the 1 s data remaining within that hour met or exceeded 75% data capture. As a*

*consequence, maximum j(NO₂)↓ presented in this study is an underestimate (7-22% based on comparison to the spectroradiometer during calibration)."*

[Figure]

**Figure AC1:** The distribution of 1-second downwelling dome measurements by month and hour-of-day that were flagged for topping out (>9 V) in 2019. These comprised 1.4% in total of raw data collected.

**2) Missing data in June**

It is important to note that data was unavailable in June was not a result of the topping out issue of the filter radiometer, but because the instrument was relocated to the University of Manchester for calibration between the 13-25[th] of that month.

The manuscript text and captions for Figs. 3 and 6 (Figs. 2 and 5 in the revised manuscript, respectively) have been updated to reflect this more clearly, as follows:

*Figure 2: "…The j(NO₂) measurements reported are the hourly mean of summed down- and up-welling components; the missing data in June are when the filter radiometer was removed from the site for calibration."*

*Figure 5: "…as measured by the filter radiometer at Auchencorth Moss for the year of study. The filter radiometer was removed from the site for calibration between 5 and 26 June 2019…."*

**3) Effect of topping-out on Fig. 5 and $p(OH)$ values**

We agree with the reviewer that the filtering may have resulted in an underestimate where the downwelling measurements exceed the topping-out threshold, particularly during midday hours or where broken cloud conditions resulted in short-term enhancements in j(NO₂). We have provided a quantification for this underestimate (7-22%; line 146, submitted manuscript)

using the calibration comparison to the spectroradiometer in June at the University of Manchester.

As mentioned in part 1 of this comments response, hourly averages of the filter radiometer measurements were only calculated if data capture in the hour exceeded 75%, once all data that topped out had been removed. As is demonstrated in Figure AC1, these mostly occur during the middle of the day. In total, this data capture threshold resulted in 3.5%, 6% and <1% of hourly average measurements not being included in spring, summer and autumn diurnal cycles, respectively. No hours were affected in the winter months. This means that a small fraction of spring and summer data could be underestimated by up to 22% during hours of full daylight, for both $p(OH)_{O_3}$ and $p(OH)_{HONO}$ rates. However since MDAF is determined separately for each hour of data, both values will be underestimated by the same quantity, resulting in the overall trends being comparable to those in the submitted manuscript.

We have revised the manuscript to emphasise that rates of $p(OH)$ from photolysis of $O_3$, HONO and $HNO_3$ discussed are estimates, due to this potential source of error limiting the upper values. Quantification of uncertainties are discussed in response to the reviewer's comments on Section 2.4.1.

Section 3.5: *"The maximum rates reached for both photolysis reactions presented here are underestimated, owing to the exclusion of the largest measurements of j(NO₂) because of the downwelling dome of filter radiometer periodically topping-out (Sect. 2.2). As these excluded measurements occur primarily during midday hours in the spring and summer months, this underestimation is likely to impact production rates of OH from the photolysis of O₃ more than from HONO. However, in this dataset the topping-out only affected 6% of summertime hours. If it is assumed all of these values are underestimated by the upper limit expected (22%; Sect. 2.2), the maximum rate of $p(OH)_{O_3}$ would increase by less than 2x10⁴ radicals cm⁻³ s⁻¹. Consequently, the overall trends observed in Fig. 8 would remain the same. Uncertainty in the seasonal estimates of $p(OH)_{HONO}$ are driven by that of measured concentrations being near the MARGA detection limit, observed in Fig. 8 with and without MDAF applied to j(HONO). For $p(OH)_{O_3}$, uncertainty from concentration measurements is much lower, but when MDAF is applied a significant amount of uncertainty is inherited (see Sect. 3.3). This combination results in less distinction between MDAF adjusted $p(OH)_{O_3}$ and $p(OH)_{HONO}$ during the full day in autumn and midday hours in spring, where these rates are comparable."*

*"Table 3. Estimated mean production rates of OH from the photolysis of O₃, HONO and HNO₃ during each meteorological season, using j-values directly output from the cloud-free TUV model and those adjusted using the MDAF metric. Means are calculated using only daytime measurements, and are reported with corresponding uncertainties discussed in Sect. 3.3. The % decrement in production rate for the MDAF data is the average decrement across the season."*

We discuss the impact of this topping-out on Fig. 5 in response to the reviewer's comments on Section 3.2 below.

*Section 2.3: Model approach:* The use of an annual mean value of ozone columns as a TUV model input (line 162) is an unnecessary simplification. The ozone column is a major factor influencing j(O1D) and j(HNO3) which is not covered by the j(NO2) measurements and is therefore not included in the MDAFs. Ozone columns are highly variable, especially during spring time in Europe and are apparently available for the site on a daily basis (or from satellite observations).

We agree with hindsight that the inclusion of the ozone column (and temperature) should have been undertaken in a more complex but realistic approach. To address this, and similar comments from Reviewer 2, we have rerun the TUV model to include measured $O_3$ column and ambient temperature as input. The temperatures used are the daily average of those already presented in the manuscript (Figure 3, submitted manuscript, now Figure 2, revised manuscript). Daily average ozone column measurements were compared between the Dobson photometer measurements at Lerwick in the Shetland Islands (Defra, 2020), and the NOAA OMI satellite data measured over Auchencorth Moss (NOAA, 2020). Good agreement was determined between these ($R^2 = 0.75$), so we opted for the satellite data due it measuring at the correct latitude and longitude, and its higher rate of data capture. These changes have been reflected in the manuscript text, figures and other associated data.

Section 2.3: *"Daily average ozone column measurements were accessed via the OMI satellite (NOAA, 2020b), and calculated for air temperature from measurements made on site (Table 1). Days where $O_3$ column measurements were missing used the measurement from the following day."*

Figure 2: *"…in this study. Satellite $O_3$ column measurements over Auchencorth Moss used in model input (NOAA, 2020b) are shown as daily values, with missing data replaced by the measurement from the following day. The j(NO$_2$) measurements…"*

Overall, the use of the TUV model in this paper is not intended to perfectly predict actinic flux or the *j*-value at any given location, but demonstrate the use of applying MDAF approaches to basic models, to account for local meteorology changes that impact atmospheric chemistry which are difficult to replicate in models.

*Section 2.3: Model approach:* Modelled j(O1D) should also take into account ambient temperature which mainly affects the quantum yields of O(1D). The temperature effect was considered for ClNO2 (mentioned later in line 206). It's unclear why it was neglected for the more important j(O1D).

We thank the reviewer for this comment, and appreciate the impact temperature can have on the quantum yield of O($^1$D) from the photolysis of $O_3$. This was not discussed in the submitted manuscript because the $\Phi$(O$^1$D) values used by the TUV model include a temperature dependence (Matsumi *et al.*, 2002) applicable to all temperatures measured during this study (267-302 K), that is currently recommended by IUPAC (Atkinson *et al.*, 2004). On the other hand, the temperature dependence of $\sigma$(ClNO$_2$) reported by Ghosh *et al.* (2012) was included in the IUPAC recommended preferred values (line 168, submitted manuscript), but not in the TUV model. Consequently there are large differences to the model output for ClNO$_2$, as we discussed in our submitted manuscript.

We have addressed the inclusion of temperature in the model input in our response to the previous comment.

*Section 2.3: Model approach:* It is also not explained why MDAFs were derived in the first place for an unimportant precursor like HNO3 and an unmeasured precursor like ClNO2 but not for a species like HCHO, a well-known and important primary radical source in this case of HO2.

The focus of the lead author's research is oxidised nitrogen chemistry, therefore focus has been placed on $NO_y$ effects. There are a significant number of organic species which are radical sources, including important precursors like HCHO, but these are not the focus of this study. However, to address this, we have now expanded the discussion in Section 3.6 to include the potential application of this metric for other photolysis reactions at the site, including potential caveats for this, and some possible solutions.

Section 3.6: *"MDAF application is not limited to the j-values presented in this study. There are a significant number of organic species which are radical sources, including other important $HO_x$ radical precursors like HCHO, where this method could easily be implemented. The TUV model, for example, has the capacity to calculate over 110 separate photolysis reactions. The primary shortcoming of the MDAF method is that using a $j(NO_2)$-derived MDAF metric means measurements lack full spectral overlap with photolysis wavelengths of some important species like $O_3$ (Sect. 3.3), contributing some uncertainty. However this can easily be minimised by use of other molecule specific radiometers (e.g. $j(O^1D)$ filter radiometer), or measurement of actinic flux across a wider wavelength range with a spectroradiometer. The latter sacrifices temporal resolution and some ease of measurement for the ability to quantify the specific j-value required for each individual study (where σ and Φ are available)."*

We acknowledge that $HNO_3$ is not an important precursor for OH radicals, although this has not yet been quantified at Auchencorth Moss. It was included in this work because $HNO_3$ is part of the suite of long-term measurements at the site. However, following comments from both Reviewer 1 and 2, we now exclude from the revised manuscript the presentation of rates of Cl production ($p(Cl)_{ClNO_2}$) from the photolysis of estimated concentrations of $ClNO_2$.

*Section 2.4.1: Adjustment factor:* The quality of the applied adjustment factors (MDAF) remains unclear and is not investigated even though it is a key parameter of this work. For example, the calculated clear sky-photolysis rate coefficients j(NO2) and j(O1D) will generally not correlate well because of the different spectral regions where the photolysis takes place which results in different dependencies on solar elevation. In addition j(O1D) is affected by temperature and ozone columns while j(NO2) is not (or very little). The uncertainty of the MDAF should be carefully estimated for each photolysis rate coefficient dependent on solar elevation and atmospheric conditions including cloud effects. Without such sensitivity studies the uncertainties of the adjusted j-values remain unclear, as well as the benefits of the whole approach.

We thank the reviewer for this response and acknowledge that we should have included a discussion of error around the application of MDAF in the submitted manuscript. In response to this, we have compared our originally $j(NO_2)$-derived MDAF, to a version using $j(O^1D)$. For this we used the spectroradiometer measurements of actinic flux made at the University of Manchester to determine $j(O^1D)$, and TUV model runs for this location and time. The model set-up was the same as it was for Auchencorth Moss, apart from the changes made to location-based data (e.g. lat/long, altitude, $O_3$ column, temperature etc).

The Figure AC2 below shows the results of this comparison: with our $j(NO_2)$ filter radiometer-derived MDAF against MDAF calculated using spectroradiometer-derived $j(NO_2)$ and $j(O^1D)$. As anticipated, agreement between $j(NO_2)$-derived MDAF's are good (grey triangles), while there is a positive bias to $j(O^1D)$-derived MDAF (mean of 23%) (coloured circles). The bias increases with solar zenith angle (SZA), where the largest difference to $j(NO_2)$-derived MDAF (69%) occurs at high SZA (>79°). This is likely attributable to the differences in scattering between the two different spectral regions of $j(NO_2)$ and $j(O^1D)$ as a result of both aerosol and cloud effects. This figure (Fig. AC2) is now included in the revised manuscript (as Fig. 6), and the caption used is the same as given here.

[Figure]

**Figure AC2:** "*Comparison between $j(NO_2)$ filter radiometer-derived MDAF and MDAF calculated using spectroradiometer-derived $j(NO_2)$ and $j(O^1D)$, at the University of Manchester (13-25 June 2019). MDAF derived using $j(NO_2)$ measurements are represented by dark grey triangles, and $j(O^1D)$ are circles coloured by the solar zenith angle (SZA), with linear regressions in dark grey and red, respectively. The 1:1 line is included in dashed grey for comparison.*"

Section 2.4: "*An MDAF metric was also determined for $j(O^1D)$, in a calculation analogous to Eq. (3), using spectroradiometer measurements of actinic flux made during calibration at the University of Manchester, $\sigma(O_3)$ from Daumont et al. (1992) and $\Phi(O^1D)$ from Matsumi et al. (2002) in Eq. (2). The TUV model was run with the same input data sources as Section 2.3, except air temperature was as measured at Manchester Airport (NOAA, 2020a). The model utilises the same values of $\Phi(O^1D)$ but sources of $\sigma(O_3)$ differ, with the model using data from Malicet et al. (1995). There is generally good agreement between this data (<2-3% difference at room temperature; Burkholder et al., 2019).*"

These derivations have been used to estimate the uncertainty in applying MDAF to other species $j$-values, propagated through where MDAF is applied (e.g. determination of $p(OH)$ rates). Since the bias between $j(NO_2)$ and $j(O^1D)$ MDAF derivations is highly variable with SZA, the bias between these two variables has been divided into 5° SZA bins. An average bias has been determined for each SZA bin, and used as the uncertainty in MDAF for that SZA bin, when applied to calculations involving $O_3$ photolysis. The same procedure is used for the photolysis of all other species investigated, except the error in MDAF is quantified from the difference between $j(NO_2)$ measurement methods used to derive each version of MDAF (spectro- vs. filter radiometer). This has been illustrated for calculation of $p(OH)$ rates in Fig. AC3 below (Fig. 8 of revised manuscript). The same figure caption text is used in the revised manuscript.

[Figure]

**Figure AC3:** "*Diurnal variations in seasonally-averaged hourly mean $p(OH)$ rates from $O_3$ and HONO photolysis. Dotted lines show rates calculated using j-values directly from the cloud-free TUV model, while solid lines show rates where j-values were first corrected by the MDAF metric, as described in the text. Shading represents the propagated uncertainty discussed in Sect. 3.3.*"

"*Section 3.3: MDAF derivations*

*The MDAF values calculated as per Eq. (3) from the measured and modelled j(NO₂) at Auchencorth Moss were largest during the sunrise and sunset hours, where SZA exceeded 80°. This is presumed to be a combined result of the model predicting incredibly small values of j(NO₂) during these hours, and radiation measured simultaneously by both domes of the filter radiometer, due to the direction of incoming radiation at high SZAs.*

*Figure 6 provides a comparison between MDAF derived from filter radiometer measurements of j(NO₂), and that calculated using j(NO₂) and j(O¹D) measured by the spectroradiometer at the University of Manchester (Sect. 2.4). There is strong agreement between $j(NO_2)$ MDAF derivations ($R^2 = 0.97$), with a mean difference of 6.9%, while there is a positive bias between $j(NO_2)$ filter radiometer and $j(O^1D)$ spectroradiometer derived MDAF values ($R^2 = 0.87$; mean = 23%). This bias is driven by increasing SZA (Fig. 6) with good agreement at low SZA (30−40°), reaching a maximum at lowest levels of solar elevation (69% at 87° SZA). This positive bias is attributed to the lack of significant overlap in spectral ranges where the photolysis of $O_3$ and $NO_2$ occur, as the wavelengths are scattered differently in the presence of atmospheric variables (e.g. clouds and aerosols). Consequently, the application of a $j(NO_2)$-derived MDAF to modelled $j(O^1D)$ results in the potential for a large level of uncertainty. Given that SZA is a driving factor in this uncertainty, the different MDAF derivations were divided into 5° SZA bins. In each bin, the mean bias from the filter radiometer-derived MDAF was quantified, and used to estimate the uncertainty of MDAF in further calculations in which that species and SZA range are used. Uncertainty in $j(O^1D)$ ranges from 8% (30−35° and 35−40° bins) to 43% (85−90°). This is unique to $j(O^1D)$, as the photolytic spectral range of $NO_2$ overlaps significantly with that of HONO (Kraus and Hofzumahaus, 1998) and $ClNO_2$ (Ghosh et al., 2012). Uncertainty in these overlapping regions ranges from ≤5% (all bins between 30−70°) to 19% at high SZA (85−90°). Overall, uncertainty is lowest where photolytic spectral regions overlap with that of measured $j(NO_2)$, and at low SZA. In order to minimise this uncertainty in future work, MDAF should be applied to modelled j-values that photolyse in a similar wavelength region to the species measured."*

*Section 2.4.3 Production rate of Cl atoms:* ClNO2 was not measured at the site (line 211). There is no reason to assume that ClNO2 concentrations are similar to those measured in Weybourne unless at least NOx and O3 levels are comparable (not shown), let alone to consider a constant mean concentration or a seasonably independent mean diurnal concentration. Consequently, the results presented later in Sect. 3.5 are pure speculation. j(ClNO2) should of course be considered as an important photolysis rate coefficient but the attempt to apply it for the site without ClNO2 measurements is pointless (see below).

We accept the reviewer's comment on this section of our manuscript, and it has now been removed. Further information regarding this is included in response to the reviewer's comments on Section 3.4 and 3.5.

*Section 3.2 Filter radiometer measurements:* Because of the limitations of the measurement technique as discussed above (Sect. 2.2), the current j(NO2) dataset does not seem suitable to derive an empirical relationship between j(NO2) and solar shortwave irradiance. Fig. 5 clearly shows that data points above the instrument's limit are missing. Moreover, as noted, some of the 1h averages shown may be biased low. So the agreement with the parametrization by Trebs et al. may in fact be better than implied by the blue line. The authors seem to recognize that because they reproduce the empirical formulas from previous studies in the text but not their own.

Figure 5 in the submitted manuscript (now Figure 4) presenting the different parameterisations between $j(NO_2)\downarrow$ and $G$ was only intended to demonstrate that the data collected by the filter radiometer in our study is in line with literature relationships, rather than imply that the current work was an updated equation for future use. The nature of the bias introduced from the 1.4% of topped-out data is discussed above (response to comments on Sect. 2.2). The missing data has affected this relationship, and as the reviewer correctly identified, is the reason we have not included our own parameterised relationship. The instrument will be renovated to improve its performance for future data collection. However this does not invalidate our approach, and our demonstration of a method that the reviewer has stated above to be a sound idea.

We have revised the text in Section 3.2 of the manuscript to reflect this, and removed the empirical formulae which could be interpreted as a comparison of equations, as picked up by the reviewer.

Section 3.2: *"A direct comparison between the co-located downwelling filter radiometer and G measurements for the duration of the study is presented in Fig. 4. The relationship between the two measurements is linear until $G\approx450$ W m$^{-2}$ above which a slight curvature is observed. A quadratic fit to this data yields predicted values of $j(NO_2)$ that lie between two previously published parameterisations between these two measurements: lower than Trebs et al. (2009) for values of G >90 W m$^{-2}$, but higher than Bahe et al. (1980) until the larger downwelling $j(NO_2)$ measured during the year. Trebs et al. used co-located G and spectro- or filter radiometer measurements from 9 sites worldwide to derive an empirical relationship between them. They show differences of up to 50% between their results, and the linear relationship proposed by Bahe et al. (1980). Trebs et al. suggest that one possible reason for the lack of curvature observed by Bahe et al. was that they did not include measurements at SZAs <30° However Auchencorth Moss is at a higher latitude than any of these sites, with a minimum SZA of ~32° in June. Curvature of the plot of $j(NO_2)$ against G is still observed, although there is still a moderate amount of scatter."*

*Due to the topping-out of the downwelling dome of the filter radiometer (discussed in Sect. 2.2), the data presented from this study is likely to be an underestimate; particularly at higher G, where measurements above the instruments limit have been excluded. Despite this, Fig. 4 demonstrates that the downwelling dome of the filter radiometer can account for changes in local conditions as well as the pyranometer, in accordance with other literature at an hourly resolution."*

*Section 3.2 Filter radiometer measurements:* The employed pyranometer also requires more attention. The SKS 1110 is not a standard pyranometer. It has a fairly limited spectral response range much more narrow than the solar range and is probably calibrated by the manufacturer by comparison with a reference pyranometer in the field. According to Coyle et al., 2019, the instrument was running for more than 20 years at the site without a follow-up calibration. Such a calibration was suggested by Coyle et al., 2019. Was it made and implemented in the current work?).

The SKS1110 is a "secondary standard grade" sensor commonly used in agro/environmental research as it provides a relatively large signal output without amplification, and is stable and reliable. The range is similar to other equivalent pyranometers (LI-COR LI-200R, Apogee Instruments SP-110 and SP-230; see links in References below). It was monitored alongside a new sensor by UKCEH for 18 months at Auchencorth Moss in 2016/17, then returned to the manufacturer and calibrated on 20 April 2018. The pyranometer was within 5% uncertainty of

a WMO Secondary Standard Pyranometer, traceable to the World Ratiometric Reference. This paper uses calibrated pyranometer data.

Section 2.1: *"Long-term instrumentation deployed at Auchencorth Moss utilised in this study is detailed in Table 1. Solar global irradiance (G) measurements were calibrated in April 2018, and certified to be within 5% uncertainty of a traceable WMO Secondary Standard Pyranometer."*

*Section 3.3 Estimates of j(HONO) and j(ClNO2):* The discussion of the empirical relationship between j(NO2) and j(HONO) by Kraus et al. is misleading. The fact that this older parametrization gives higher values than the TUV model is not a result related with the j(NO2) measurements of the present study. It's just a property of the TUV model which used more recent recommendations of absorption cross sections and quantum yields. Qualitatively the discrepancy is explainable because Kraus et al. used recommendations from 1998: smaller quantum yields for the NO2 photolysis and greater absorption cross sections for HONO. What is more important for this study is that j(NO2) and j(HONO) from TUV probably correlate linearly in good approximation which justifies the use of the MDAF concept for j(HONO).

We agree that the comparison to this older parameterisation highlights how the relationship can change when the molecular-specific values are updated. We also agree that a bit more explanation is needed, and have adjusted the text to include the different molecular-specific values used in each calculation to better reflect the point that Reviewer 1 makes, and ensure it is not misleading.

Section 3.4: *"…consistent over all seasons. This is likely to be attributed to the updated molecule-specific parameters used by the TUV model, c.f. Kraus and Hofzumahaus (1998) for both $NO_2$ and HONO. Absorption cross-sections ($\sigma$) are different for both species. $\sigma$(HONO) varies the most, with an agreement of better than 10% between the JPL recommended values used in the TUV model (Sander et al., 2011) and the results of Bongartz et al. (1994) used by Kraus and Hofzumahaus (Burkholder et al., 2019). For $\sigma(NO_2)$, results of Harwood and Jones (1994) used by Kraus and Hofzumahaus are 6−8% below most other studies (Burkholder et al., 2019), including that of Vandaele et al. (1998) and Mérienne et al. (1995) used in this study (in the TUV model and spectroradiometer calculations in Sect. 2.2, respectively). The quantum yield for HONO photolysis is assumed to be unity in both calculations, but $\Phi(NO_2)$ used by the TUV model (and in spectroradiometer calculations; Troe, 2000) are on average 4% greater than the combination of Gardner et al. (1987) and Roehl et al. (1994) used by Kraus and Hofzumahaus."*

*Section 3.3 Estimates of j(HONO) and j(ClNO2):* For j(ClNO2) a similar problem arises: The result that TUV overestimates j(ClNO2) has again nothing to do with the j(NO2) measurements. It's simply because the implemented absorption cross sections are obviously outdated as noted in Sect. 2.3 of the paper. How much greater the values are is secondary for the current study but it could of course be mentioned in the text as a friendly reminder for the TUV developers to update their input.

The reviewers' observation is correct. We feel this is a valuable addition to the science, as both TUV developers and other atmospheric chemistry modellers should understand when parameters are in need of review. We have added a sentence to the manuscript to emphasise this, and made the discussion of these parameterisations more general.

Section 3.4: *"Overall, the TUV model yields the greatest annual mean $j(ClNO_2)$ values (Fig. 7, middle panel), exceeding the others by 16-26%, with the maximum difference at midday. Both Young et al. (2014) and Riedel et al. (2014) demonstrate reasonable agreement to the updated $j(ClNO_2)$ results, suggesting this difference is in part contributed by outdated $\sigma(ClNO_2)$ used in the TUV model. The inclusion of $j(O^1D)$ in the parameterisation by Young et al. appears to result in smaller predicted $j(ClNO_2)$ values c.f. Riedel et al., particularly in the morning hours (04:00-10:00 UTC). However, these results are within the uncertainty associated with applying the MDAF metric (Sect. 3.3), and consequently distinctions between these cannot be made with great certainty."*

*Section 3.3 Estimates of j(HONO) and j(ClNO2):* A discussion of j(O1D) is completely missing here. The points raised in Sect. 2.3 and 2.4.1 should be considered and a reasonable uncertainty estimate should be derived for all photolysis rate coefficients. The MDAF concept is far more complicated and error-prone in particular for j(O1D) than currently implied in the paper.

The comments here have been taken on board by the authors. We did not include $j(O^1D)$ estimations in Section 3.3 of the submitted manuscript as this was not also compared to parameterisations that estimate $j(O^1D)$. However we agree with the comments made by the reviewer, and acknowledge that not including this discussion elsewhere in the manuscript was inappropriate.

In our revisions we have renamed this section and revised the text to include $j(O^1D)$, and included a new panel in Fig. 7 (Fig. AC4 below, the caption of which is also used in the revised manuscript) to show its annual average diurnal cycle where MDAF has been applied, with associated uncertainty. This uncertainty attributable to using a $j(NO_2)$-derived MDAF on modelled $j(O^1D)$ has been derived in response to the reviewers comments on Section 2.4.1 and 3.4/3.5, and in the new section in the revised manuscript (Sect. 3.3: MDAF derivations).

*"Section 3.4: Estimates of j(HONO), $j(ClNO_2)$ and $j(O^1D)$*

*The annual average diurnal cycle of $j(O^1D)$ output by the TUV model is presented in the bottom panel of Fig. 7, alongside the uncertainty associated with applying MDAF to these results, determined for each SZA bin before averaging (Sect. 3.3). The uncertainty is greatest during sunrise (04:00-07:00 UTC) and sunset hours (18:00-20:00 UTC), at 35-42%, but results in largest range of $j(O^1D)$ during midday hours (~15% uncertainty). This is almost twice as much uncertainty as attributed to either $j(HONO)$ or $j(ClNO_2)$."*

[Figure]

**Figure AC4:** "***Top:*** *Annual average diurnal cycle of j(HONO) at Auchencorth Moss calculated by the TUV model and using the Kraus and Hofzumahaus (1998) parameterisation.* ***Middle:*** *Annual average diurnal cycle of j(ClNO₂) estimates for Auchencorth Moss, using the direct TUV model output and updated j(ClNO₂) values (accounting for daily average temperature), and parameterisations presented in* Riedel *et al.* (2014) *and Young* et al. (2014). ***Bottom:*** *Annual average cycle of j(O¹D) at Auchencorth Moss, calculated by the TUV model. All values of j(HONO), j(ClNO₂) and j(O¹D) are the annual mean of each hour with the MDAF metric applied, to account for local influence on solar fluxes. Shading for all data represents the uncertainty determined from MDAF derivation methods, discussed in Sect. 3.3.*"

*Section 3.4 and 3.5: Production rate of OH radicals / Production rate of Cl atoms:* That HNO3 photolysis is insignificant (line 290) as an OH precursor is by no means new information and could have been estimated from clear sky summer maximum OH production rates alone. The impression is that HNO3 was included in this study only because it was measured.

We agree that the demonstration here that $HNO_3$ photolysis is a minor OH production pathway is not new; however given this study was looking at $NO_y$, the authors feel that it is not irrelevant to include. As mentioned in our response to previous comments, this parameter has not yet been estimated at Auchencorth Moss, and since its concentration is already an established long-term measurement at this site, it has been included, alongside its propagated uncertainty (including that from concentration measurements and MDAF derivation).

*Section 3.4 and 3.5: Production rate of OH radicals / Production rate of Cl atoms:* The explanations in lines 297-305 regarding the different seasonal dependencies of p(OH) from O3 and HONO are not reproducible. Cloud effects may be a factor but the main seasonal difference is that j(O1D) and j(HONO) depend differently on solar elevation. The diurnal variations of p(OH) from O3 are probably not the same in summer and winter (line 303). The authors will notice that when they multiply the winter data with 25 (line 304).

We thank the reviewer for pointing out this inaccuracy. We have revised this paragraph in the manuscript to more accurately reflect the likely explanation for these differences, attributing them to solar elevation *c.f.* the presence of overcast cloud. The mention of the diurnal variation of $p(OH)_{O_3}$ was not intended to imply that the diurnal cycles were completely identical in summer and winter, as we are aware this is not the case. We aimed to illustrate the point that for all seasons, the diurnal cycle of $p(OH)_{O_3}$ has broadly the same bell shape, tracking the solar intensity (*c.f.* $p(OH)_{HONO}$ which is larger in the morning). This has been rectified in our revised manuscript.

Section 3.5: *"…Despite this, the greatest rates of $p(OH)$ are still observed in the summer, with seasonal average peak in $p(OH)_{O_3}$ at 13:00 (UTC), at approximately 1.7 x10[6] radicals cm[-3] s[-1], around half the rate observed in Melbourne, Australia in March (Ryan et al., 2018). These rates are comparable when a MDAF is not applied, due to similar concentrations of $O_3$ (~54 μg m[-3] average in Australia). In the same season at Auchencorth Moss, $p(OH)_{HONO}$ reaches a maximum, approximately half that of $p(OH)_{O_3}$, at 09:00 (UTC; ~5 hours after sunrise). As concentrations of HONO are higher in Melbourne than at Auchencorth Moss, $p(OH)_{HONO}$ is around 8 times greater. A similar pattern is observed in spring (Fig. 8), where $p(OH)_{HONO}$ is greater in the early morning hours (~04:00-09:00 UTC), before $p(OH)_{O_3}$ increases to a comparable level in the middle of the day, closely following the diurnal cycle of light intensity and peaking when shorter wavelengths of light are more prevalent. This broadly similar diurnal pattern of $p(OH)_{O_3}$ is observed in all seasons, but a difference of ~20 is observed between peak $p(OH)_{O_3}$ in summer and winter, despite similar concentrations of $O_3$ (61.0 μg m[-3] and 57.1 μg m[-3], respectively). This is a consequence of the change in solar elevation between the two seasons. During this study, the minimum SZA (complementary to the solar elevation angle) in winter is 64° c.f. 32° in summer, meaning the sun is further from the zenith and lower in the sky. This results in a longer path length of light through the atmosphere, and increases the probability that light will be scattered or absorbed (e.g. by clouds or aerosols) before reaching the point of observation. Consequently the shorter wavelengths of light necessary for $O_3$ photolysis (≤ 340 nm) readily scatter before reaching ground-level, c.f. the longer wavelengths*

*required for HONO photolysis (305 ≤ λ ≤ 420 nm). For this reason, over all daylight hours in winter, $p(OH)_{HONO}$ consistently exceeds $p(OH)_{O_3}$ by a factor of ~5."*

*Section 3.4 and 3.5: Production rate of OH radicals / Production rate of Cl atoms:* The comparison of the obtained p(OH) with those from a single study in Australia is arbitrary. That HONO photolysis is a significant or even dominant OH precursor (at ground level) is well known and was discussed in many well designed studies before. The current paper gives little new insight as the quality of the HONO measurements remains unclear as well as local NOx levels that could classify the results.

We agree with the reviewer that HONO as a source of OH at ground level is not a new finding, and although there are numerous studies showing this, these rates have not yet been quantified at Auchencorth Moss. Long-term HONO measurements are spatially infrequent (particularly *c.f.* $O_3$), making this an important dataset (unique in the UK) with which to test our understanding of this photolysis reaction, and the application of MDAF. The comparison to the recent work by Ryan *et al.* (2018) in Melbourne, Australia was deliberately included as context for our results, since these authors explicitly present $p(OH)_{HONO}$ and $p(OH)_{O_3}$ rates calculated using the same methodology as this study. We have reduced this discussion to a short comparison between peak rates reached (see revision in quote from Section 3.5 in response to previous comment).

We appreciate the suggestion made by the reviewer to include further discussion of the concentration measurements used. We have revised the manuscript to include further discussion of the quality of the gaseous MARGA measurements used (see quote from Section 3.1 below), as well as annual average $NO_x$ concentrations measured at Auchencorth Moss, from a manuscript currently in preparation about $NO_x$ measurements at the site.

Section 3.1: *"Annual average concentrations of $NO_2$ and NO were 2.84 and 0.18 µg m⁻³, respectively (Thermo Scientific 42C NO-$NO_2$-$NO_x$ analyser, ThermoFisher, USA; Cowan et al., in prep.)."*

Cowan, N., Leeson, S. R., Twigg, M. M., Jones, M. R., Harvey, D., Simmons, I., M., C., and Braban, C.: Assessing the bias of molybdenum catalytic conversion in the measurement of $NO_2$ in rural air quality networks, in preparation.

Section 3.1: *"Median HONO concentrations show far less diurnal and seasonal variation, but a greater interquartile range of concentrations for each hour in the summer. Larger concentration episodes were observed in the spring and summer months, particularly July, reflected by the largest seasonal mean and median concentrations occurring in summer (0.065 and 0.096 µg m⁻³, respectively).*

*MARGA measurements are predicted to overestimate HONO concentrations, as a result of heterogeneous chemistry of $NO_2$ on inlet surfaces and the wet rotating denuder (WRD). This potential overestimate has not yet been quantified at Auchencorth Moss, but the nearest geographical comparison (~8.5 km NE at Easter Bush) showed that a Gradient of Aerosols and Gases Online Registrator (GRAEGOR, ECN, NL; with the same sampling and analytical method as MARGA) recorded measurements ~6% higher than a co-located Long Path Absorption Photometer (LOPAP), an artefact-free HONO measurement method (Ramsay et al., 2018). In contrast, MARGA measurements in Melpitz, Germany (Stieger et al., 2018), show this artefact to be 58% (Ren et al., 2020). The length of the inlet plays a large role in the*

*artefact potential, with the longer inlet at Melpitz providing a greater surface area for heterogeneous reactions of NO$_2$ to occur. The inlet length at Auchencorth Moss lies between that used at Easter Bush and Melpitz, so it is anticipated that the potential interference in concentration is likely to be between these two values.*

*Mean and median HNO$_3$ concentrations were largest in the spring (0.16 and 0.10 μg m$^{-3}$, respectively), due to a large peak in mid-April reaching a maximum of 1.45 μg m$^{-3}$. Daytime HNO$_3$ is likely to be underestimated, as it is expected that there will be losses of HNO$_3$ to the inlet, although the magnitude of this loss at Auchencorth Moss has not yet been quantified. A study using a similar set-up to Auchencorth Moss reported that HNO$_3$ was ~50% less than a standard filter method for sampling HNO$_3$ (Makkonen et al., 2014). Nocturnal MARGA measurements of HNO$_3$, on the other hand, have been shown to include positive interference from the heterogeneous hydrolysis of N$_2$O$_5$ (Phillips et al., 2013). However since nocturnal filter radiometer measurements are below the detection limit at night, this interference is not predicted to affect $p(OH)_{HNO_3}$ rates presented in this study."*

*Section 3.4 and 3.5: Production rate of OH radicals / Production rate of Cl atoms:* As was noted already in Section 2.4.3 the discussion of Cl production rates based on devised diurnal concentrations is too speculative. So the lengthy discussion on possible diurnal or seasonal variations of Cl production at the site is not productive. That ClNO2 photolysis can lead to an increased VOC oxidation in particular during morning hours has been discussed in a number of previous studies, however based on ClNO2 measurements. ClNO2 concentrations are typically extremely variable. Compared to that potential uncertainties related with photolysis rate coefficients seem to be a minor problem. The final sentence in line 345 intended to emphasize the (potential) impact of Cl atoms for the radical budget at the site is also misleading. The fact that Cl atoms react faster with many VOCs than OH is secondary. Ultimately they both react mostly with VOCs and what counts are the production rates.

We would like to thank the reviewer for this input. Although we were aware of the limitation of using estimated ClNO$_2$ concentrations we had originally decided that inclusion of a $p(Cl)_{ClNO_2}$ rate estimation would provide useful additional comparison with the other radical production rates derived for the Auchencorth site. However, following comments from both Reviewer 1 and 2, we accept that there is too much conjecture in these estimations of Cl radical production. Therefore this section and its associated figures (Sect. 3.5 and Figs. 2 and 10, submitted manuscript) have been removed from the revised manuscript.

*Section 3.4 and 3.5: Production rate of OH radicals / Production rate of Cl atoms:* Overall the sections on the radical production rates are not very conclusive. The production rates are important parameters but they are just a small fraction of production and destruction rates during daytime that make up tropospheric photochemistry. So the absolute numbers are difficult to interpret unless you run a box model for comparison or you manage to extract these parameters from an operational model for this location. Moreover, this work is not about the concentrations of the precursors and whether or not they are represented correctly in current models. If you would just discuss the photolysis rate coefficients, a very similar picture would emerge regarding the importance of local effects on radiation. In accordance with that, the whole introduction is concerned with radiation, not with radical production rates.

The purpose of including these radical production rates was to demonstrate the impact that MDAF can have on further calculations of photochemical parameters, as well as a potential useful application of long-term $j$-value measurements at a site (in this case, Auchencorth Moss). The absolute quantities of $p(OH)_{O_3}$ and $p(OH)_{HONO}$ may be difficult to interpret alone, but are useful for comparison against one another and similar studies where these values are necessary.

Although the production rates compared here (from HONO and $O_3$ photolysis) comprise a small fraction of the overall tropospheric photochemistry taking place at the site, they are fundamentally important as sources of local OH radical formation. It is therefore crucial that these are determined accurately in a chemical reaction scheme to prevent propagation of large errors in further calculations. As we show in our study, application of MDAF reduced both $p(OH)$ rates determined for Auchencorth Moss by ~40% throughout the year compared to a basic cloud-free model representation, which is a very significant. Inclusion of this metric provides a minimal contribution to computational cost *c.f.* an extensive radiative transfer scheme that also needs accurate cloud cover, making it a useful alternative. We have revised the manuscript to improve clarity of these points.

Section 2.5: *"The impact of a locally-driven MDAF is demonstrated here through the context of OH radical production rates at Auchencorth Moss, from the photolysis of sources in existing long-term measurements at the site including $O_3$ and two $NO_y$ species: HONO and $HNO_3$."*

Section 3.6: "*Accurate j-values are integral to accurate assessment of air quality, in particular to photolytic production of radicals and subsequent secondary pollutants which negatively impact both human and environmental health, including tropospheric $O_3$ and particulate matter. Radiometer measurements will consequently have immense value in supporting atmospheric chemistry measurement and modelling. A demonstration of the application of MDAF at Auchencorth Moss (Sect. 3.5) shows that where concentrations of relevant atmospheric precursors are co-located with long-term filter radiometer measurements, other metrics like the production rate of OH radicals can be quantified. These can provide additional routes for model validation and assessment, as these parameters are fundamental in model representations of tropospheric photochemistry.*"

*Section 3.6 and 3.7: Implications of the MDAF metric / Conclusions:* The paragraph following line 351, "In contrast, Sommariva et al. report: : :." is confusing: MCM parametrizations, different altitudes, and different seasons all mixed together. I don't think this arbitrary citation is necessary to emphasize the need for locally measured photolysis rate coefficients for a proper analysis of field data.

We agree with the reviewer about the lack of clarity in our original phrasing. However we do believe this citation is relevant as it provides a view of the discrepancies that can occur when measurement and model locations are not the same, in contrast to this study. We have reworded this section of the manuscript to improve clarity.

Section 3.6: *"This work, together with the recent study by Sommariva et al. (2020), highlights the importance of coincident measurement and model locations when generating a local MDAF metric."*

*Section 3.6 and 3.7: Implications of the MDAF metric / Conclusions:* The paragraphs line 364-370 and 387-390 where "long-term multi-site radiometer measurements" (line 364) are suggested, apparently draft a main conclusion of the study which is also implied in the title. However, I don't think such an investment would be justified based on the current analysis.

The investment justification is an opinion rather than a critical comment on the science presented, and the reviewer may have a different opinion to the authors.

*Section 3.6 and 3.7: Implications of the MDAF metric / Conclusions:* The authors themselves show that a fairly compact empirical relationship exists between j(NO2) and solar shortwave radiation (Trebs et al., 2009). Even though this parameterization is not perfect, it will nevertheless cover a large fraction of the most important local cloud effects.

Although the empirical relationship is good, it is also valuable to understand the accurate details of local atmospheric chemistry, particularly as pollutants such as $NO_2$ are changing with policy or air quality management interventions. Operation of filter radiometers are straightforward to set-up and operate, and would provide a valuable dataset at sites in existing long-term networks, such as the EMEP network.

*Section 3.6 and 3.7: Implications of the MDAF metric / Conclusions:* As the authors also state, pyranometers are already used within existing networks and have a high degree of standardization. So to "provide a higher spatial density of model-measurement jvalue comparison points in the UK" (line 387) a first step could be to use these already available long term data from many stations to evaluate the model data and to investigate the impact of potential shortcomings of model-predicted j-values. If it then turns out that the j(NO2) derived from the pyranometer data are not accurate enough for this purpose you would have a point. But I doubt this will be the case.

The authors thank the reviewer for their comments and thoughts on the future work.

For "supersites" like Auchencorth Moss, having a long-term on-site set of radiometer measurements would greatly improve model representations of the site for the many campaigns or studies that take place there on account of the quantity of instrumentation co-located there. Similar sites in existing air quality networks would also benefit from this addition. Pyranometer measurements can be used in local photochemistry modelling, however these are approximations with varying degrees of accuracy. To understand or validate model chemical mechanisms at given sites, *j*-values measurements are better. A UK-wide study using the pyranometer network would be very interesting.

**Specific comments**

Line 5: "In this paper, locally representative photolysis rate constants (j-values) for these molecules are shown to be critical for quantifying and understanding the rate of radical production in a local atmosphere." Is this a new insight? j-values are routinely measured since decades during photochemical campaigns for exactly that reason.

Although *j*-values have been measured in campaigns for decades, long-term measurements are much rarer and provide the volume of data to address chemistry in a changing physical and chemical atmosphere. We agree that this is not a new insight as a standalone phrase, and so have reworded the abstract accordingly.

Abstract: "*Locally representative photolysis rate constants (j-values) for these molecules are critical for quantifying and understanding the rate of radical production in a local atmosphere.*"

Line 55: "Cl radicals are extremely reactive…, Cl has a significant effect on local tropospheric oxidation." I think this statement is misleading. Low Cl concentrations may be a result of the high reactivity of Cl but the importance of Cl depends on its production rate. That it's more reactive than OH is unimportant (moreover, OH is regenerated which makes it the key oxidant).

We had included this statement to emphasise that Cl should not be ignored when considering the oxidising capacity of the local atmosphere. However, we agree with the points made by the reviewer, so to avoid any further potential misinterpretation, we have revised this part of the manuscript.

Section 1: "*Along with OH, Cl radicals are extremely reactive with VOCs, where rate constants can be up to an order of magnitude greater than equivalent reactions with OH (Monks, 2005; Young et al., 2012). Therefore, even in low abundances, Cl plays an important role in local tropospheric oxidation (Riedel et al., 2014; Bannan et al., 2015) but this varies with its production rate, as Cl is not as easily regenerated as OH.*"

Line 102: "…when measured AOD and NO2 concentrations are included in model inputs, actinic flux is reasonably well predicted…" It should be clarified that this statement probably refers to clear-sky conditions.

We agree with this comment, and have revised the statement referencing to Palancar *et al.* (2013) accordingly.

Section 1: "*Palancar et al., (2013) demonstrate that when parameters measured within the planetary boundary layer (PBL) are included in model input (including AOD, $O_3$ column, single scattering albedo and $NO_2$ concentrations), actinic flux is reasonably well predicted on cloudless days, and the main source of uncertainty is then attributable to σ and Φ.*"

Line 134: Give a reference where the Bentham instrument and its uncertainties are described or provide more information.

Publications where the Bentham spectroradiometer used for calibration were also deployed are now referenced, and the overall measurement uncertainty has been detailed and cited in the text.

Section 2.2: *"Output signals from the filter radiometer were calibrated at the University of Manchester against a Bentham DTM300 scanning spectroradiometer (Webb et al., 2002a; Thiel et al., 2008) between 13 and 25 June 2019. This mid-summer period was selected to provide calibration over the maximum range of ambient incident radiation. The direction of the filter radiometer was turned 180° mid-way through this period, in order to calibrate each dome separately. Overall uncertainty in the spectroradiometer measurements was ±5%, the sum in quadrature of uncertainties in instrumental parameters (including cosine and actinic head responses), measured wavelength and general setup."*

Line 170: Why is Section 2.4 entitled "Theoretical calculations"? In this Section you combine measured data with those from the TUV model.

This section was primarily titled as theoretical as it includes the production rate of Cl atoms from $ClNO_2$, which at present has not been measured at the site, making all estimates of production rates theoretical. This section header has now been removed completely (see response to previous Section 3.5), and all subsections within it have been changed to individual sections.

Line 223: "UK seasons are defined…" Is this definition useful with regard to radiation measurements?

We included this clarification to ensure that all readers would be able to easily identify exactly which data were included in each of the seasonal averages presented.

Line 247, Eq. (9): Convert to units used in this work.

This comment is no longer applicable, as this equation has been removed from the revised manuscript (see response to comments on Section 3.2).

Line 258: "The MDAF values…was largest during sunrise and sunset hours" This is explainable by an instrumental artefact. At low sun direct radiation can strike the upper and the lower dome simultaneously unless your "artificial horizon" is large enough to prevent this. For the common Metcon instruments it is not.

We are aware of these artefact potentials in total filter radiometer measurements from 4-π instruments. However we believe, as we mention in the submitted manuscript, that the large MDAF values are largely due to extremely small non-zero values during these hours, where $j$-values are expected to be small and filter radiometer measurements are very close to the detection limit. Following a similar comment by the Reviewer 2, we have stated this potential

interference in the filter radiometer section of the methods (Section 2.2), and at this later point in the manuscript, which is in Section 3.3.

Section 2.2: *"This restriction is not perfect as increased sensitivity can be observed near the horizon in 4-π systems due to contribution from the opposite dome. Hofzumahaus et al. (2002) show that this bias is partially compensated by the reduced sensitivity in individual 2-π inlets, resulting in a maximum overestimation of their actinic flux of 4% (between 0-12 km altitude)."*

Section 3.3: *"The MDAF values calculated as per Eq. (3) from the measured and modelled $j(NO_2)$ at Auchencorth Moss were largest during the sunrise and sunset hours, where SZA exceeded 80°. This is presumed to be a combined result of the model predicting incredibly small values of $j(NO_2)$ during these hours, and radiation measured simultaneously by both domes of the filter radiometer, due to the direction of incoming radiation at high SZAs."*

Line 260: "..the adjustment factor was greatest in the morning and steadily decreases throughout the day until sunset". Is there an explanation for this behaviour? The mean diurnal variations of j(HONO) and j(ClNO2) in Fig. 7 don't show this.

The decrease referred to was from the annual average MDAF values, and was a small difference between sunrise and sunset. We do not have an explanation for this, so to avoid any misleading interpretations, we have removed this section from the manuscript.

Line 294: I do not see the 71% decrease in Fig. 8. It looks more like the overall 40% decrease stated in line 293.

The stated 71% decrease referred to the hourly values, where the maximum reduction in $j(O^1D)$ through the year was 71% in the summertime measurements, not the average decrease of ~40%. We appreciate the reviewer asking for clarification of this, and have updated the sentence accordingly.

Section 3.5: *"The largest decrease in measurements as a result of MDAF application occurs in the summer months (~50%; see Table 3), as the cloud-free model conditions are an inaccurate representation of summertime conditions at Auchencorth Moss."*

Line 385ff: "The enhanced contribution…" Please reconsider this statement taking into account the different dependencies of j(O1D) and j(HONO) on solar elevation. The clear-sky TUV results will show the same effect.

We agree with the comment made by the reviewer, and have addressed this in our response to comments on Sections 3.4/3.5. We have revised the text in the conclusion to reflect this.

Section 4: "*The enhanced contribution of HONO photolysis in the colder months is a consequence of the lower solar elevation in winter (minimum SZA of 64° c.f. 32° in summer), reducing the shorter wavelengths available for $O_3$ photolysis, relatively more than the longer wavelengths contributing to HONO photolysis."*

Figure 3: The gap in the j(NO2) data should be noted in the caption.

The relocation of the filter radiometer to Manchester for the calibration (see response to Section 2.2) has been specified in the caption to Fig. 2 in the revised manuscript (Fig. 3, submitted manuscript).

*Figure 2: "…The j(NO$_2$) measurements reported are the hourly mean of summed down- and up-welling components; the missing data in June are when the filter radiometer was removed from the site for calibration."*

Figure 6: What can be recognized in this figure are daily maximum values rather than hourly data. A correlation would perhaps be more insightful. The gap in the data requires an explanation.

Plotting a correlation would be a useful interpretation; however we plotted this figure (Fig. 5 in revised manuscript) to illustrate the temporal variation in both measured and modelled j(NO$_2$) throughout the year, and to show occasions where measured j(NO$_2$) exceeded the cloud-free TUV model estimates. The main example of this is in early February where upwelling contributed more due to the larger surface albedo, shown by the inset Phenocam image. These aims would be harder to present as easily in a correlation plot. The gap in data, due to the relocation of the filter radiometer (see response to Section 2.2), has been addressed in the revised caption for this figure.

*Figure 5: "…as measured by the filter radiometer at Auchencorth Moss for the year of study. The filter radiometer was removed from the site for calibration between 5 and 26 June 2019…."*

Figure 7, 8, 9: The 95% confidence intervals are virtually invisible and it's unclear what they represent.

Following the inclusion of Section 3.3 (revised manuscript), we have revised the figures in the revised manuscript. The shading now represents the total error in each of these values, propagated from errors in concentration measurements, and error in MDAF application (see response to Section 2.4.1).

**References**

Atkinson, R., Baulch, D. L., Cox, R. A., Crowley, J. N., Hampson, R. F., Hynes, R. G., Jenkin, M. E., Rossi, M. J. and Troe, J.: Evaluated kinetic and photochemical data for atmospheric chemistry: Volume I - gas phase reactions of $O_x$, $HO_x$, $NO_x$ and $SO_x$ species, *Atmos. Chem. Phys.,* 4, 1461–1738, doi:10.5194/acp-4-1461-2004, 2004.

Apogee Instruments, *Pyranometer Owner's Manual for Models SP-110 and SP-230*, available at: https://www.apogeeinstruments.com/content/SP-110-manual.pdf, (accessed: 3 September 2020), 2020.

Bahe, F. C., Schurath, U. and Becker, K. H.: The frequency of $NO_2$ photolysis at ground level, as recorded by a continuous actinometer, *Atmos. Environ.,* 14, 711–718, 1980.

Bannan, T. J., Booth, A. M., Bacak, A., Muller, J. B. A., Leather, K. E., Le Breton, M., Jones, B., Young, D., Coe, H., Allan, J., Visser, S., Slowik, J. G., Furger, M., Prévôt, A. S. H., Lee, J., Dunmore, R. E., Hopkins, J. R., Hamilton, J. F., Lewis, A. C., Whalley, L. K., Sharp, T., Stone, D., Heard, D. E., Fleming, Z. L., Leigh, R., Shallcross, D. E. and Percival, C. J.: The first UK measurements of nitryl chloride using a chemical ionization mass spectrometer in central London in the summer of 2012, and an investigation of the role of Cl atom oxidation, *J. Geophys. Res. Atmos.,* 120, 5638–5657, doi:10.1002/2014JD022629, 2015.

Bongartz, A., Kames, J. and Schurath, U.: Experimental Determination of HONO Mass Accommodation Coefficients Using Two Different Techniques, *J. Atmos. Chem.,* 18, 149–169, doi:10.1007/BF00696812, 1994.

Burkholder, J. B., Abbatt, J. P. D., Cappa, C., Dibble, T. S., Kolb, C. E., Orkin, V. L., Wilmouth, D. M., Sander, S. P., Barker, J. R., Crounse, J. D., Huie, R. E., Kurylo, M. J., Percival, C. J. and Wine, P. H.: Chemical Kinetics and Photochemical Data for Use in Atmospheric Studies. Jet. Propul. Lab., Pasedena, CA, available at: http://jpldataeval.jpl.nasa.gov/, 2019.

Cowan, N., Leeson, S. R., Twigg, M. M., Jones, M. R., Harvey, D., Simmons, I., M., C., and Braban, C.: Assessing the bias of molybdenum catalytic conversion in the measurement of $NO_2$ in rural air quality networks, *in preparation*.

Daumont, D., Brion, J., Charbonnier, J. and Malicet, J.: Ozone UV spectroscopy I: Absorption Cross-Sections at Room Temperature, *J. Atmos. Chem.,* 15, 145–155, doi:10.1007/BF00053756, 1992.

Defra: *Department for Environment, Food and Rural Affairs - Stratospheric Ozone Data,* available at: https://uk-air.defra.gov.uk/data/ozone-data, (accessed: 1 March 2020), 2020.

Gardner, E. P., Sperry, P. D. and Calvert, J. G.: Primary Quantum Yields of $NO_2$ Photodissociation, *J. Geophys. Res. Atmos.,* 92, 6642–6652, doi:10.1029/JD092iD06p06642, 1987.

Ghosh, B., Papanastasiou, D. K., Talukdar, R. K., Roberts, J. M. and Burkholder, J. B.: Nitryl Chloride ($ClNO_2$): UV/Vis Absorption Spectrum between 210 and 296 K and $O(^3P)$ Quantum Yield at 193 and 248 nm, *J. Phys. Chem. A*, 116, 5796–5805, doi:10.1021/jp207389y, 2012.

Harwood, M. H. and Jones, R. L.: Temperature dependent ultraviolet-visible absorption cross sections of $NO_2$ and $N_2O_4$: Low-temperature measurements of the equilibrium constant for $2NO_2 \rightleftharpoons N_2O_4$, *J. Geophys. Res. Atmos.*, 99, 22,955-22,964, doi:10.1029/94jd01635, 1994.

Hofzumahaus, A., Kraus, A., Kylling, A. and Zerefos, C. S.: Solar actinic radiation (280-420 nm) in the cloud-free troposphere between ground and 12 km altitude: Measurements and model results, *J. Geophys. Res. Atmos.,* 107, doi:10.1029/2001JD900142, 2002.

Kraus, A. and Hofzumahaus, A.: Field Measurements of Atmospheric Photolysis Frequencies for $O_3$, $NO_2$, HCHO, $CH_3CHO$, $H_2O_2$ and HONO by UV Spectroradiometry, *J. Atmos. Chem.*, 31, 161–180, doi:10.1023/A:1005888220949, 1998.

LI-COR, *LI-COR LI-200R Pyranometer*, available at: https://www.licor.com/env/products/light/pyranometer.html, (accessed: 3 September 2020), 2020.

Makkonen, U., Virkkula, A., Hellén, H., Hemmilä, M., Sund, J., Äijälä, M., Ehn, M., Junninen, H., Keronen, P., Petäjä, T., Worsnop, D., Kulmala, M. and Hakola, H.: Semi-continuous gas and inorganic aerosol measurements at a boreal forest site: seasonal and diurnal cycles of $NH_3$, HONO and $HNO_3$, *Boreal Environ. Res.*, 19, 311–328, 2014.

Malicet, J., Daumont, D., Charbonnier, J., Parisse, C., Chakir, A. and Brion, J.: Ozone UV spectroscopy. II. Absorption Cross-Sections and Temperature Dependence, *J. Atmos. Chem.*, 21, 263–273, doi:10.1007/BF00696758, 1995.

Matsumi, Y., Comes, F. J., Hancock, G., Hofzumahaus, A., Hynes, A. J., Kawasaki, M. and Ravishankara, A. R.: Quantum yields for production of $O(^1D)$ in the ultraviolet photolysis of ozone: Recommendation based on evaluation of laboratory data, *J. Geophys. Res. Atmos.*, 107, 4024, doi:10.1029/2001jd000510, 2002.

Mérienne, M. F., Jenouvrier, A. and Coquart, B.: The $NO_2$ Absorption Spectrum. I: Absorption Cross-Sections at Ambient Temperature in the 300-500 nm Region, *J. Atmos. Chem.,* 20, 281–297, doi:10.1007/BF00694498, 1995.

Monks, P. S.: Gas-phase radical chemistry in the troposphere, *Chem. Soc. Rev.,* 34, 376–395, doi:10.1039/b307982c, 2005.

NOAA: *National Oceanic and Atmospheric Administration - Integrated Surface Database (NOAA-ISD)*, available at: https://www.ncdc.noaa.gov/isd, 2020.

NOAA: *National Oceanic and Atmospheric Administration Earth System Research Laboratory Global Monitoring Devision (NOAA-ESRL-GMD) OMI Satellite Level 3e daily averaged ozone data gridded at 0.25 x 0.25 degrees*, available at: https://www.esrl.noaa.gov/gmd/grad/neubrew/SatO3DataTimeSeries.jsp, (accessed: 1 August 2020), 2020.

Palancar, G. G., Lefer, B. L., Hall, S. R., Shaw, W. J., Corr, C. A., Herndon, S. C., Slusser, J. R. and Madronich, S.: Effect of aerosols and $NO_2$ concentration on ultraviolet actinic flux near Mexico City during MILAGRO: measurements and model calculations, *Atmos. Chem. Phys.,* 13, 1011–1022, doi:10.5194/acp-13-1011-2013, 2013.

Phillips, G. J., Makkonen, U., Schuster, G., Sobanski, N., Hakola, H. and Crowley, J. N.: The detection of nocturnal $N_2O_5$ as $HNO_3$ by alkali-and aqueous-denuder techniques, *Atmos. Meas. Tech.*, 6, 231–237, doi:10.5194/amt-6-231-2013, 2013.

Ramsay, R., Marco, C. F. Di, Heal, M. R., Twigg, M. M., Cowan, N., Jones, M. R., Leeson, S. R., Bloss, W. J., Kramer, L. J., Crilley, L., Sörgel, M., Andreae, M. and Nemitz, E.: Surface – atmosphere exchange of water – soluble gases and aerosols above agricultural grassland pre – and postfertilisation, *Atmos. Chem. Phys.*, 18, 16953–16978, doi:10.5194/acp-18-16953-2018, 2018.

Ren, Y., Stieger, B., Spindler, G., Grosselin, B., Mellouki, A., Tuch, T., Wiedensohler, A. and Herrmann, H.: Role of the dew water on the ground surface in HONO distribution: a case measurement in Melpitz, *Atmos. Chem. Phys. Discuss.,* 2020.

Riedel, T. P., Wolfe, G. M., Danas, K. T., Gilman, J. B., Kuster, W. C., Bon, D. M., Vlasenko, A., Li, S.-M., Williams, E. J., Lerner, B. M., Veres, P. R., Roberts, J. M., Holloway, J. S., Lefer, B., Brown, S. S. and Thornton, J. A.: An MCM modeling study of nitryl chloride ($ClNO_2$) impacts on oxidation, ozone production and nitrogen oxide partitioning in polluted continental outflow, *Atmos. Chem. Phys.*, 14, 3789–3800, doi:10.5194/acp-14-3789-2014, 2014.

Roehl, C. M., Orlando, J. J., Tyndall, G. S., Shetter, R. E., Vázquez, G. J., Cantrell, C. A. and Calvert, J. G.: Temperature Dependence of the Quantum Yields for the Photolysis of $NO_2$ near the Dissociation Limit, *J. Phys. Chem.,* 98, 7837–7843, doi:10.1021/j100083a015, 1994.

Ryan, R. G., Rhodes, S., Tully, M., Wilson, S., Jones, N., Frieß, U. and Schofield, R.: Daytime HONO, $NO_2$ and aerosol distributions from MAX-DOAS observations in Melbourne, *Atmos. Chem. Phys.,* 18, 13969–13985, doi:10.5194/acp-2018-409, 2018.

Sander, S. P., Friedl, R. R., Barker, J. R., Golden, D. M., Kurylo, M. J., Wine, P. H., Abbatt, J. P. D., Burkholder, J. B., Kolb, C. E., Moortgat, G. K., Huie, R. E. and Orkin, V. L.: Chemical Kinetics and Photochemical Data for Use in Atmospheric Studies, Evaluation Number 17. JPL Publication 10-6, Pasadena CA, 2011.

Skye Instruments, *Skye SKS 1110 Pyranometer*, available at: http://www.skyeinstruments.info/index_htm_files/Pyranometer.pdf, (accessed: 3 September 2020), 2020.

Sommariva, R., Cox, S., Martin, C., Borońska, K., Young, J., Jimack, P. K., Pilling, M. J., Matthaios, V. N., Nelson, B. S., Newland, M. J., Panagi, M., Bloss, W. J., Monks, P. S. and Rickard, A. R.: AtChem (version 1), an open-source box model for the Master Chemical Mechanism, *Geosci. Model Dev.*, 13, 169–183, doi:10.5194/gmd-13-169-2020, 2020.

Stieger, B., Spindler, G., Fahlbusch, B., Müller, K., Grüner, A., Poulain, L., Thöni, L., Seitler, E., Wallasch, & M., Herrmann, H., Chem, J. A. and De, H.: Measurements of $PM_{10}$ ions and trace gases with the online system MARGA at the research station Melpitz in Germany – A five-year study, *J. Atmos. Chem.*, 75, 33–70, doi:10.1007/s10874-017-9361-0, 2018.

Thiel, S., Ammannato, L., Bais, A., Bandy, B., Blumthaler, M., Bohn, B., Engelsen, O., Gobbi, G. P., Gröbner, J., Jäkel, E., Junkermann, W., Kazadzis, S., Kift, R., Kjeldstad, B., Kouremeti, N., Kylling, A., Mayer, B., Monks, P. S., Reeves, C. E., Schallhart, B., Scheirer, R., Schmidt, S., Schmitt, R., Schreder, J., Silbernagl, R., Topaloglou, C., Thorseth, T. M., Webb, A. R., Wendisch, M. and Werle, P.: Influence of clouds on the spectral actinic flux density in the lower troposphere (INSPECTRO): Overview of the field campaigns, *Atmos. Chem. Phys.*, 8, 1789–1812, doi:10.5194/acp-8-1789-2008, 2008.

Trebs, I., Bohn, B., Ammann, C., Rummel, U., Blumthaler, M., Königstedt, R., Meixner, F. X., Fan, S. and Andreae, M. O.: Relationship between the $NO_2$ photolysis frequency and the solar global irradiance, *Atmos. Meas. Tech.*, 2, 725–739, doi:10.5194/amt-2-725-2009, 2009.

Troe, J.: Are Primary Quantum Yields of $NO_2$ Photolysis at $\lambda \leq 398$ nm Smaller than Unity?, *Zeitschrift für Phys. Chemie*, 214, 573–581, doi:10.1524/zpch.2000.214.5.573, 2000.

Vandaele, A. C., Hermans, C., Simon, P. C., Carleer, M., Colin, R., Fally, S., Mérienne, M. F., Jenouvrier, A. and Coquart, B.: Measurements of the $NO_2$ absorption cross-section from 42,000 $cm^{-1}$ to 10,000 $cm^{-1}$ (238-1000 nm) at 220 K and 294 K, *J. Quant. Spectrosc. Radiat. Transfer*, 59, 171–184, doi:10.1016/S0022-4073(97)00168-4, 1998.

Webb, A. R., Bais, A. F., Blumthaler, M., Gobbi, G.-P., Kylling, A., Schmitt, R., Thiel, S., Barnaba, F., Danielsen, T., Junkermann, W., Kazantzidis, A., Kelly, P., Kift, R., Liberti, G. L., Misslbeck, M., Schallhart, B., Schreder, J. and Topaloglou, C.: Measuring Spectral Actinic Flux and Irradiance: Experimental Results from the Actinic Flux Determination from Measurements of Irradiance (ADMIRA) Project, *J. Atmos. Ocean. Technol.*, 19, 1049–1062, doi:10.1175/1520-0426(2002)019<1049:MSAFAI>2.0.CO;2, 2002.

Young, C. J., Washenfelder, R. A., Edwards, P. M., Parrish, D. D., Gilman, J. B., Kuster, W. C., Mielke, L. H., Osthoff, H. D., Tsai, C., Pikelnaya, O., Stutz, J., Veres, P. R., Roberts, J. M., Griffith, S., Dusanter, S., Stevens, P. S., Flynn, J., Grossberg, N., Lefer, B., Holloway, J. S., Peischl, J., Ryerson, T. B., Atlas, E. L., Blake, D. R. and Brown, S. S.: Chlorine as a primary radical: evaluation of methods to understand its role in initiation of oxidative cycles, *Atmos. Chem. Phys.*, 14, 3427–3440, doi:10.5194/acp-14-3427-2014, 2014.

Young, C. J., Washenfelder, R. A., Roberts, J. M., Mielke, L. H., Osthoff, H. D., Tsai, C., Pikelnaya, O., Stutz, J., Veres, P. R., Cochran, A. K., Vandenboer, T. C., Flynn, J., Grossberg, N., Haman, C. L., Lefer, B., Stark, H., Graus, M., De Gouw, J., Gilman, J. B., Kuster, W. C. and Brown, S. S.: Vertically resolved measurements of nighttime radical reservoirs in Los Angeles and their contribution to the urban radical budget, *Environ. Sci. Technol.,* 46, 10965–10973, doi:10.1021/es302206a, 2012.

---

## Author Comment (AC2) · 24 Sep 2020

**amt-2020-219: Use of filter radiometer measurements to derive local photolysis rates and for future monitoring network application (Walker *et al.*)**

**Responses to Reviewer 2**

The authors thank Reviewer 1 for their time spent conducting a careful review of our paper, and providing constructive comments. We hope that our comments and corresponding revisions to the original manuscript are satisfactory.

The major revisions to the manuscript after comments from both Reviewer 1 and 2 include the following.

- Re-running of the TUV model to include measurements of daily average ozone column and ambient temperature (see our response to comments on Section 2.3).
- Estimates of  $p(Cl)_{ClNO_2}$  rates have been excluded and Section 3.5 (*"Production rate of Cl atom radials"*, submitted manuscript) has been removed (see response to Section 2.4.3 and 3.4/3.5).
- A new section has been introduced (*"Section 3.3: MDAF derivations"*, revised manuscript) to detail methods of determining the uncertainty arising from applying a *j*(NO2)-derived MDAF, as we present here, to other species *j*-values (see response to Section 2.4.1).

Our responses to individual points raised by the review are listed below. The suggestions and comments made by the reviewer are listed in **black font**, and our responses highlighted in blue, with any relevant manuscript changes indicated in *blue italic*. We have also amended the subsectional structuring, and some other text (e.g. abstract) of our manuscript as a consequence the revisions we describe above and below.

**General comments**

The work ratios long-term filter radiometer jNO2 measurements to a cloud-free radiative transfer model (TUV) to calculate a measurement-driven adjustment factor (MDAF). Modeled jO3, jHONO, jCINO2 and jHNO3 are then multiplied (presumably) by the MDAF correction factor to determine the environmentally impacted local values. Combining these rates with local measurements, the authors calculate radical products and find a significant reduction in OH compared with the cloud-free modeled products. The authors suggest that such local radiation measurements would significantly improve chemical model calculations of important species. The local impact of clouds, aerosols and changing albedo on photolysis rates and the resulting impacts on radical chemistry require additional validation in models. The long-term dataset discussed provides the opportunity to examine a range of conditions to test model sensitivity to changes in local conditions and the incentive to set up additional jNO2 measurement locations. The paper is relevant but lacks rigor in some areas. With revisions and additional discussion the paper would be worthy of publication.

We are pleased that the reviewer has identified the relevance of this method. The authors would like to thank the reviewer for the overall positive review, and their constructive comments. We have taken forward most of the reviewer's suggestions, and think that the adjustments we have made have significantly improved the manuscript and addressed the scientific issues raised.

**Specific comments**

Line 13: 40% lower OH production rate compared to what? Presumably compared to the cloud-free model. Best for clarity to be explicit. Some discussion of cloud-free vs cloud resolving models would be useful.

The reviewers' interpretation of this statement is correct, and we agree with their statement. We have revised the text in the manuscript accordingly.

Abstract: "The MDAF resulted in these rates being ~40% lower than the cloud-free model output over the entire year."

Line 35: I suggest an alternative PAN reference:

Singh, H. B., Herlth, D., O'Hara, D., Zahnle, K., Bradshaw, J. D., Sandholm, S. T., Talbot, R., Crutzen, P. J., and Kanakidou, M.: Relationship of Peroxyacetyl nitrate to active and total odd nitrogen at northern high latitudes: Influence of reservoir species on NOx and O3, J. Geophys. Res, 97, 16523–16530, 1992.

The reference for PAN contributing to regional pollution has been updated to include the suggested citation, and a modelling study which also determined the importance of PAN in redistributing  $NO_x$ .

Section 1: "...and atmospheric reservoirs of NO2 such as peroxyacetyl nitrate (PAN), which contribute to long-range and regional pollution (Singh et al., 1992; Moxim et al., 1996)."

**Line 70: Many molecules include a pressure dependence. I suggest adding pressure (p) to equations.**

We agree that pressure is another variable influencing molecule-specific parameters used to determine the *j*-values, and Eq. (2) in the revised manuscript has been updated to reflect this. We did not originally include pressure in Eq. (2) as the IUPAC and JPL-recommended absorption cross-section and quantum yield values used in our calculations did not include quantified pressure dependencies. For example, Sander *et al.* (2006) state that the effects of variation in NO2 absorption cross-sections with total pressure is only observed when measured at spectral resolutions higher than 0.01 nm, which is greater than that used in our calculations (1 nm). Similarly, Burkholder *et al.* (2019) present evidence that no pressure dependence has been experimentally observed for the O3 absorption cross-section.

Section 1: Eq. (2):  $j = \int_{\lambda_1}^{\lambda_2} \sigma(\lambda, T, p) \phi(\lambda, T, p) F(\lambda) d\lambda$

Line 72: The definition attributed to Madronich is confusing. Actinic flux is more simply the spherically integrated radiation through a sphere or the radiant energy density incident on a unit spherical surface.

We thank the reviewer for highlighting this issue with the wording of our definition of actinic flux. We have simplified the text in the manuscript to convey the message more clearly.

Section 1: "The actinic flux (F) is the spherically integrated radiation available to a molecule in the atmosphere, which upon absorption results in photodissociation (Madronich, 1987)."

Line 78: Here and throughout the paper, the authors refer to applying or using the MDAF. However, I don't see the "application" is ever explicitly defined. I am left to presume (perhaps obviously) that MDAF is a simple multiplication factor. That should be stated here.

There is no mention of MDAF at or around line 78; instead we believe that this comment was intended for line 178, where we agree with the reviewer, and thank them for noticing this omission. The process of applying MDAF, as correctly presumed by the reviewer, has been included in more detail in Section 2.4.

Section 2.4: "Adjustment factors like MDAF can be derived from other spatio-temporally coincident measured and modelled j-values, such as j(O1D). The product of MDAF and the initially modelled (cloud-free) j-values are referred to as "adjusted j-values". These factors have only occasionally been applied both spatially and temporally (Stone et al., 2012; Bannan et al., 2015)..."

Line 86: This seems like an odd line noting that jNO2 instruments have "limited potential to estimate the photolysis frequencies of other atmospheric species." Isn't that the point of this paper? I think you are saying that they cannot \*directly\* measure the other species without MDAF or a similar correction.

The interpretation presented by the reviewer is the point which we intended to make. To provide more clarity, we have rephrased this sentence to include the word "directly".

Section 1: "However, they remain reliant on absolute calibrations to directly quantify j-values from recorded voltages. Each instrument is only applicable for its specified photolysis reaction, and cannot be used to directly estimate the photolysis frequencies of other atmospheric species. Filter radiometers most commonly measure j(NO2) and j(O1D)."

Line 103: This is an incorrect oversimplification. The Palancar data was screened for clouds and included PBL heights in addition to AOD and NO2 concentrations.

We thank the reviewer for spotting this. We have added further clarification to the citation of Palancar *et al.* (2013) to better reflect the study and avoid possible misinterpretation.

Section 1: "Palancar et al. (2013) demonstrate that when parameters measured within the planetary boundary layer (PBL) are included in model input (including AOD, O3 column, single scattering albedo and NO2 concentrations), actinic flux is reasonably well predicted on cloudless days, and the main source of uncertainty is then attributable to  $\sigma$  and  $\phi$ ."

Line 127: The difficulties of a 4 pi spectrometer should be discussed. In particular, the upwelling can be significantly influenced by the support tower, any nearby equipment and the local albedo (e.g. a bush or rock below the tower). This can add significant uncertainty, particularly when the upwelling is large (e.g. snow). Also, the errors in the two 2-pi optics are particularly large near the horizion. This is noted throughout the literature (e.g. Hofzumahaus, et al, (2002) doi:10.1029/2001JD900142 and references therein). See the comment at lines 258-61.

We thank the reviewer for this constructive comment. As the Auchencorth Moss site comprises lots of instrumentation, these potential sources of uncertainty and bias in the filter radiometer measurements were carefully considered during set-up. We agree that many of the issues associated with the filter radiometer placement can introduce error to the measurements, but due to logistical issues (e.g. the necessity of a mast and a finite cable length) are sadly unavoidable. To compensate we have included these features in the description of the filter radiometer in Section 2.2. We appreciate the reviewer drawing our attention to the Hofzumahaus *et al.* (2002) study, and we now include a small discussion of these principles in this section. Errors associated directly with the inlet optics of our filter radiometer are discussed with the response to the reviewer's comments at lines 258-61.

Section 2.2: "The 4- $\pi$  filter radiometer (Metcon, Meteorologie Consult GmbH, Germany) was mounted ~3 m above the ground, recording measurements at 1 s time resolution for a full year (21 November 2018 – 20 November 2019). The position of the instrument was carefully considered to minimise potential sources of interference from the site. Situated at the outskirts of the site, the filter radiometer was directed away from all other objects. Its supporting mast had a matte black coating to reduce any reflection that wouldn't otherwise occur. The ground cover beneath the instrument was largely long grasses, where features that could increase surface albedo (e.g. snow) were quite evenly distributed. The closest change in these features (approx. 5 m behind the supporting mast) is a wooden slatted path covered in black non-slip mats. This could contribute some uncertainty to upwelling measurements, particularly during conditions with a large surface albedo, but is deemed minimal due to its distance and how it is often obscured by vegetation growth. The inlet optic of each dome is designed to have a near-uniform angular response through use of a guartz diffusor (Bohn et al., 2004). Each optical inlet is surrounded by a light shield to provide an "artificial horizon", restricting the field-of-view for each dome to one hemisphere (Volz-Thomas et al., 1996). This restriction is not perfect as increased sensitivity can be observed near the horizon in  $4-\pi$  systems due to contribution from the opposite dome. Hofzumahaus et al. (2002) show that this bias is partially compensated by the reduced sensitivity in individual 2- $\pi$  inlets, resulting in a maximum overestimation of their actinic flux of 4% (between 0-12 km altitude)."

Lines 146-9: I recommend being explicit about the averaging for clarity. The filter radiometer is broadband and continuously measures the full jNO2 spectrum. The scanning spectrometer measures one narrow wavelength band linearly in time over the 3 minute duty cycle. Thus, the measurements are not equivalent in rapidly changing conditions.

We agree with these comments made by the reviewer. During a 3 minute scan where conditions are changeable, the bandpass measurements made by the filter radiometer would not be equivalent to the sum of the wavelengths scanned by the spectroradiometer. This is why we used the standard deviation of measured 1 s filter radiometer measurements as an

estimate of how changeable the conditions were during each scan. Any time this standard deviation in filter radiometer measurements during the relevant 3 minute interval was exceeded, that comparison point was removed from the calibration dataset. We have revised the text in the manuscript to detail this process.

Section 2.2: "During the calibration, broadband measurements made by the filter radiometer (1 s) during each spectroradiometer scan (actinic flux measured in each narrow  $\lambda$  band sequentially; 3 mins) were averaged to obtain a comparison point. Large standard deviations associated with these mean values were used to remove calibration points where actinic flux was highly variable during the scan (e.g. rapidly changing cloud cover), which would result in inconsistent conditions between  $\lambda$  bands and render an unrepresentative comparison to the filter radiometer. This range was primarily within ±5% of the mean calculated."

Line 158: Be consistent TUV 5.3 or 5.3.1 (as noted in the abstract).

We thank the reviewer for identifying this inconsistency. The TUV model is now referred to as v5.3.1 throughout the manuscript.

Line 161: The model is not technically "clear-sky" because aerosols are included. I suggest using "cloud-free" throughout.

We have changed all references of the "clear-sky model" to "cloud-free" as suggested.

Lines 163-4: List the default values for albedo and aerosols (presumably 0.10 and the Elterman (1967) continental aerosol profile).

L. Elterman UV, Visible and IR Attenuation to 50 Km, AFCRL-68-0153, Environ. Res. Papers, (No. 278) (1968), Bedford, Mass.

The values mentioned by the reviewer are correct, and are now directly cited in the revised manuscript. Following comments from both Reviewers 1 and 2, the TUV model has been rerun to include daily average ozone column and ambient temperature in the model input, in order to more accurately estimate modelled  $j(O^{1}D)$ . These alterations have also now been mentioned in Section 2.3 and in response to the reviewers' comments on line 295.

Section 2.3: "The model was set up for the location and height of the filter radiometer at Auchencorth Moss, and assumed to have cloud-free conditions. Daily average ozone column measurements were accessed via the OMI satellite (NOAA, 2020b), and calculated for air temperature from measurements made on site (Table 1). Days where O3 column measurements were missing used the measurement from the following day. Default TUV values for surface albedo (0.10) and AOD at 500 nm (Elterman, 1968) were used."

Lines 168-9: I believe default TUV uses a different cross section (Vandaele et al., 1998). According to JPL 2015, Vandaele and Mérienne are within 2-3% across their measurement spectral range.

We appreciate the reviewer highlighting this error in the manuscript. We were mistaken in this particular aspect of the TUV model based on our interpretation of the source code. The revised manuscript now includes this reference, and a comparison to the absorption cross-sections used in our calculations.

Section 2.3: "It should be noted that for j(NO2) calculations, the NO2 quantum yield used by the TUV model is the same as that used in the calibration of the filter radiometer (Troe, 2000), while the absorption cross-section uses measurements made by Vandaele et al. (1998), which differ from the values used in calibration (Mérienne et al., 1995) by 2-3% at room temperature (Burkholder et al., 2019)."

**Line 190: Describe kH2O, kN2 and kO2.**

These parameters have now been identified in the description of Eq. (6) in the revised manuscript.

Section 2.5: "In these equations,  $j(O^1D)$  is the photolysis rate constant for  $O_3$ , and f is the fraction of  $O({}^1D)$  atoms that react with water vapour to form OH, as opposed to their quenched removal by  $N_2$  and  $O_2$  molecules. Rate constants for these individual reactions between  $O({}^1D)$  and  $H_2O$ ,  $N_2$  and  $O_2$  ( $k_{H_2O}$ ,  $k_{N_2}$  and  $k_{O_2}$ , respectively) were taken from Atkinson et al. (2004). Temperature dependence was included for quenching reactions but not  $k_{H_2O}$ , as it is stated to be independent of temperature between 200-350 K."

**Line 207: Eq 2 is specific to jNO2. One option would be to make Eq 2 generic and adjust the ("in this case") text in line 75.**

Equation 2 was made specific to  $j(NO_2)$  for relevance to the context in which it is first presented in the introduction. However the reviewer correctly points out this is less applicable where it is referenced later in the manuscript. The equation (and subsequent explanation) has been updated to ensure it remains generic and relevant for all instances to which it is referred.

Section 1: Eq. (2):  $j = \int_{\lambda_1}^{\lambda_2} \sigma(\lambda, T, p) \phi(\lambda, T, p) F(\lambda) d\lambda$

This section of text (describing the updated *j*(CINO2) calculations) has been moved to Section 2.3, as Section 2.4.3 has been removed from the manuscript (described further in response to comments on line 215).

Section 2.3: "Since the dataset presented by Ghosh et al. includes the temperature dependence of  $\sigma(CINO_2)$ , values in this study were parameterised according to the daily mean temperature measured at Auchencorth Moss prior to use in Eq. (2). Henceforth,  $j(CINO_2)$  calculated with these values are referred to as "updated  $j(CINO_2)$ ". It should be noted that for  $j(NO_2)$  calculations..."

**Line 215: Discuss expected seasonal variation in CINO2 as Sommariva was summertime only.**

This would be a good addition to provide context to the  $p(Cl)_{ClNO_2}$  calculations. However following the combination of feedback we received from Reviewers 1 and 2, we have decided that this section of the manuscript should be removed due to the lack of physical measurements of ClNO2 at the Auchencorth Moss site. (We initially decided  $p(Cl)_{ClNO_2}$  would be a useful additional example of radical production routes at the site, but we now acknowledge that its inclusion contains too much conjecture.)

Line 258-61: A figure would be helpful to understand the SZA relationship to MDAF. At low sun, the angular response of the radiometer optics will contribute significantly to the MDAF. How does the filter radiometer data compare to the Bentham at low sun? In addition, the model will be particularly sensitive to the aerosols applied in TUV (Elterman). These will generally differ from the actual profile (including the PBL height). I expect the MDAF analysis is not particularly effective at very low sun (perhaps >85 deg sza). This topic should be addressed and perhaps such data needs to be excluded.

We agree with this assessment of MDAF in relation to SZA. In comparison to the Bentham spectroradiometer, we have coloured the data points in the calibration data (Fig. 1) by the SZA measured by the spectroradiometer (see Fig. AC1, below). The plot below is a magnification of the figure in the discussion manuscript so as to focus on the higher SZA (range of filter radiometer from 10 V to 3 V, and spectroradiometer from  $8 \times 10^{-3} \text{ s}^{-1}$  to  $3 \times 10^{-3} \text{ s}^{-1}$ . The caption for this figure is that now used in the revised manuscript. The maximum SZA observed for down- and upwelling domes were 91.69° and 91.34°, respectively. From this we have assumed that from this calibration, SZA measured by each dome up to 90° are within the error of its calibration.

**Figure AC1:** "Calibration of both filter radiometer optical inlets against a Bentham DTM300 spectroradiometer at the University of Manchester from 13-25 June 2019. Filter radiometer measurements (1 s) are averaged to equal the scan duration of spectroradiometer (approx. 3 mins), and data where conditions were highly variable for this period (e.g. cloud cover), have been excluded. Points are coloured by the solar zenith angle (SZA) measured by the spectroradiometer (30–91°). Relationships used for subsequent conversion of filter radiometer measurements are presented in black."

This version of Fig. 1 has magnified both axes close to the origin to demonstrate the agreement at high solar zenith angles. The version included in the revised manuscript is shown in Fig. AC3 below.

In Fig. AC2 below we have plotted the hourly calculated MDAF against the SZA, before any data was filtered out. A  $log_{10}$  scale is used to include all data points. A huge uptick in MDAF is observed at highest SZA, which was found to be primarily driven by extremely small values of *j*(NO2) calculated by the TUV model and slightly higher measurements due to actinic flux at low solar elevation being measured simultaneously by both domes (model as low as ~10-13 s-1 at times when ~10-5 s-1 was measured).

**Figure AC2:** Hourly filter radiometer  $j(NO_2)$ -derived MDAF values as a function of hourly SZA determined by the TUV model prior to data filtering, on a log10 scale. The vertical line indicates an SZA of 90°, where measurements made by the filter radiometer are within the error of calibration against the University of Manchester Bentham DTM300 spectroradiometer.

In our submitted manuscript, filter radiometer measurements below the detection limit were excluded from MDAF calculations. MDAF was also not calculated if modelled  $j(NO_2)$  was below the detection limit. This resulted in MDAF values calculated at a maximum SZA of 93°. Following this comment made by the reviewer, the approach has been changed in the revised manuscript to use the SZA calculated by the TUV model ( $R^2 \ge 0.99$  when compared to that measured by the spectroradiometer we calibrated against). The filter radiometer is accurate within the error of calibration to 90°, which also bisects the curve in Fig. AC2 before modelled values result in a fast increase in MDAF with SZA. Therefore we excluded MDAF values where SZA was >90°.

- Section 2.2: "...this mid-summer period was selected to provide calibration over the maximum range of ambient incident radiation, between solar zenith angles (SZA) of 30-91°."
- Section 2.2: "...calibration error of each dome. Overall errors for down- and upwelling domes were 13% and 12%, respectively for SZA between 30° and 91°."
- Section 2.4: "In this study at Auchencorth Moss, MDAF values were excluded if they were calculated at a SZA >90°, as these conditions are likely to result in more significant uncertainty in measurements, and were not observed during calibration."

**Figure AC3:** "Calibration of both filter radiometer optical inlet domes against a Bentham DTM300 spectroradiometer at the University of Manchester from 13–25 June 2019. Filter radiometer measurements (1 s) are averaged to equal the scan duration of spectroradiometer (approx. 3 mins), and data where conditions were highly variable for this period (e.g. cloud cover), have been excluded. Points are coloured by the solar zenith angle (SZA) measured by the spectroradiometer (30.1–91.69° downwelling; 30.1–91.34° upwelling). Relationships used for subsequent conversion of filter radiometer measurements are presented in black."

Line 287: The discussion of HNO3 is a bit odd. Either it should be given the full analysis of the other molecules or it should excluded because it is not relevant to the OH analysis. The low OH production from HNO3 is not even mentioned in the conclusion, perhaps because it is not an interesting finding of the study. If HNO3 is included, why not a list of other photolysis frequencies?

The focus of the lead author's research is oxidised nitrogen chemistry, therefore focus has been placed on  $NO_y$  effects. We acknowledge that  $HNO_3$  is not an important precursor for OH radicals, although this has not yet been quantified at Auchencorth Moss. It was included because  $HNO_3$  is part of the suite of long-term measurements at the site. We have revised the conclusion to mention  $HNO_3$ .

Section 4: "The enhanced contribution of HONO photolysis in the colder months is a consequence of the lower solar elevation in winter (minimum SZA of 64° c.f. 32° in summer), reducing the shorter wavelengths available for O3 photolysis, relatively more than the longer wavelengths contributing to HONO photolysis. In all seasons,  $p(OH)_{HNO_3}$  is negligible compared to  $p(OH)_{O_3}$  and  $p(OH)_{HONO}$ , reaching seasonal average rates 103 radicals cm-3 s-1 lower."

There are a significant number of organic species which are radical sources, including important precursors like HCHO, but these are not the focus of this study. However, to address this we have now expanded the discussion in Section 3.6 to include the potential application of this metric for other photolysis reactions at the site, including potential caveats for this, and some possible solutions (see our response to the Reviewers comments on Line 364-70).

Line 289 and Table 3: I think this must state "cloud-free TUV" for clarity.

Yes, these corrections to references to the TUV model output have been updated in the manuscript as suggested in response to the reviewer's previous comment.

Line 295: jNO2 and jO3 cover significantly different spectral ranges resulting in differing diurnal profiles and interactions with clouds and aerosols. More importantly, jO3 is highly dependent on the O3 column. Applying the MDAF to jO3 will result in significant uncertainty and I question whether it is a valid method. For more, see Lefer et al., 2003 (doi:10.1029/2002JD003171). The jCINO2 and jHONO have spectral parameters similar enough to jNO2 that MDAF is likely ok. The paper lacks any discussion of uncertainties resulting from the MDAF results for each of the molecules.

We fully agree with this comment, and a similar point raised by Reviewer 1. Consequently, we have now re-run the TUV model to include the daily average measured  $O_3$  column and ambient temperature as input. The temperatures used are the daily average of measurements already presented in the manuscript (Figure 3, submitted manuscript). Daily average ozone column measurements were compared between the Dobson photometer measurements at Lerwick in the Shetland Islands (Defra, 2020), and the NOAA OMI satellite data measured over Auchencorth Moss (NOAA, 2020b). Good agreement was determined between these ( $R^2 = 0.75$ ), so the satellite data was used due to it measuring at the correct latitude and longitude, and its higher rate of data capture. Both time series can be seen in Fig. AC4 below (Fig. 2, revised manuscript), with the same caption used in the revised manuscript.

**Section 2.3: "Daily average ozone column measurements were accessed via the OMI satellite (NOAA, 2020b), and calculated for air temperature from measurements made on site (Table 1). Days where O3 column measurements were missing used the measurement from the following day."**

Overall, the use of the TUV model in this paper is not intended to perfectly predict actinic flux or the *j*-value at any given location, but demonstrate the use of applying MDAF approaches to basic models, to account for local meteorology changes that impact atmospheric chemistry which are difficult to replicate in models.